# Inhibition of double-strand DNA-sensing cGAS ameliorates brain injury after ischemic stroke

Qian Li[1], Yuze Cao[1], Chun Dang[2], Bin Han[2], Ranran Han[2], Heping Ma[3], Junwei Hao[4,*] & Lihua Wang[1,**]

## Abstract

**Cytosolic double-stranded DNA (dsDNA) is a danger signal that is tightly monitored and sensed by nucleic acid-sensing pattern recognition receptors. We study the inflammatory cascade on dsDNA recognition and investigate the neuroprotective effect of cyclic GMP-AMP (cGAMP) synthase (cGAS) antagonist A151 and its mechanisms of neuroprotection in a mouse model of experimental stroke. Here, we found that cerebral ischemia promoted the release of dsDNA into the cytosol, where it initiated inflammatory responses by activating the cGAS. A151 effectively reduced the expression of cGAS, absent in melanoma 2 (AIM2) inflammasome, and pyroptosis-related molecules, including caspase-1, gasdermin D, IL-1β, and IL-18. Furthermore, mice treated with A151 showed a dampened immune response to stroke, with reduced counts of neutrophils, microglia, and microglial production of IL-6 and TNF-α after MCAO. Moreover, A151 administration significantly reduced infarct volume, attenuated neurodeficits, and diminished cell death. Notably, the protective effect of A151 was blocked in a microglia-specific cGAS knockout mouse. These findings offer unique perspectives on stroke pathogenesis and indicate that inhibition of cGAS could attenuate brain inflammatory burden, representing a potential therapeutic opportunity for stroke.**

**Keywords** cGAS; microglia; neuroinflammation; pyroptosis; stroke
**Subject Categories** Cardiovascular System; Immunology; Molecular Biology of Disease

## Introduction

Ischemic stroke is a devastating neurological disease worldwide, with high global burden (Mukherjee & Patil, 2011). Although the only FDA-approved drug against acute ischemic stroke, recombinant tissue plasminogen activator (rtPA), has shown clinical efficiency,

however, due to the narrow therapeutic time window, majority of patients miss the optimal opportunity for vascular recanalization (Wechsler, 2011; Fu *et al*, 2015). Brain damage induced by initial ischemia is most likely unsalvageable; however, inflammation exists over an extended period until days following ischemia, which may facilitate the secondary neural immunologic deterioration and expand cerebral infarction, thus offering many opportunities for treatment strategies (Iadecola & Anrather, 2011; Fu *et al*, 2015). After the ictus of brain ischemic attack, cytosolic double-stranded DNA (dsDNA) released by necrotic neuronal cells is a potential damage-associated molecular pattern (DAMP), the underlying mechanisms of brain inflammation on dsDNA recognition by nucleic acid-sensing cyclic GMP-AMP (cGAMP) synthase (cGAS) following stroke have not been explored to date.

cGAS has currently been defined as a key cytosolic DNA sensor (Gao *et al*, 2013; Cai *et al*, 2014). Upon binding DNA directly, cGAS catalyzes the synthesis of cyclic dinucleotide 2′–3′- cGAMP, a second messenger, which in turn interacts with and activates the stimulator of interferon genes (STING) to induce the production of type I interferons via transcription factors interferon regulatory factor 3 (Chen *et al*, 2016; Xia *et al*, 2016) and of pro-inflammatory cytokines (e.g., IL-6 and TNF-a) via transcription factors nuclear factor κB (NF-κB) (Ablasser *et al*, 2013; Cai *et al*, 2014). Recently, it was shown that cytosolic DNA detected by the cGAS-STING axis could induce a cell death program and that was associated with inflammasome activation, thus targeting cGAS signaling is likely to be a logical approach of controlling inflammatory response triggered by cytoplasm dsDNA and ameliorating the related pathology (Gaidt *et al*, 2017).

In addition, a series of studies have demonstrated that a protein complex called inflammasome absent in melanoma 2 (AIM2) plays a central role in various sterile self-DNA triggered inflammatory conditions, such as radiation-induced gastrointestinal syndrome and hematopoietic failure (Hu *et al*, 2016). AIM2 recognizes cytosolic dsDNA and directly binds it through its C-terminal HIN-200 domain, and then changes its conformation and triggers the "prion-like" polymerization of the inflammasome adaptor protein apoptosis-associated speck-like protein containing a CARD (ASC) into a filamentous helical structure, offering a multitude of activation sites for the inflammasome

1 Department of Neurology, The Second Affiliated Hospital, Harbin Medical University, Harbin, China
2 Department of Neurology, Tianjin Neurological Institute, Tianjin Medical University General Hospital, Tianjin, China
3 Department of Physiology, Emory University School of Medicine, Atlanta, GA, USA
4 Department of Neurology, Xuanwu Hospital, Capital Medical University, Beijing, China
 *Corresponding author. Tel: +86 1083 198707; E-mail: haojunwei@vip.163.com
 **Corresponding author. Tel: +86 45186 297475; E-mail: wanglh211@163.com

effector caspase-1. Consequently, caspase-1 is activated to drive the proteolytic cleavage and maturation of precursor cytokines, including pro-interleukin-1β (IL-1β) and pro-interleukin-18 (pro-IL-18), leading to the extracellular release of pro-inflammatory cytokines (Hornung *et al*, 2009). In addition to the post-translational assembly of inflammasome that is directly triggered by dsDNA, type I interferons stimulated by the cGAS-STING pathway have been shown to be capable of enhancing the AIM2 inflammasome response by promoting the expression of inflammasome platforms and substrates (Martinon *et al*, 2009; Lugrin & Martinon, 2018). Moreover, caspase-1 has been demonstrated to trigger a form of programmed cell death called pyroptosis (Broz & Dixit, 2016), which is inherently high inflammatory (Bergsbaken *et al*, 2009). Gasdermin D (GSDMD) has recently emerged as a major substrate of caspase-1 responsible for pyroptosis; of note, this inflammatory caspase-mediated program death differs from caspase-mediated apoptosis in that it causes localized cellular swelling and disrupts membrane integrity, thereby leading to the extravasation of cellular contents and inflammatory mediators into the extracellular milieu (Bergsbaken *et al*, 2009; Broz & Dixit, 2016; Aglietti & Dueber, 2017; Shi *et al*, 2017).

We are optimistic that manipulating the innate DNA-sensing signaling may be an optimistic therapeutic intervention to improve the stroke outcomes. The synthetic oligodeoxynucleotide, A151, comprised of the immunosuppressive motif TTAGGG, has recently been shown to be capable to abrogate activation of cytosolic nucleic acid-sensing cGAS and AIM2 inflammasome by binding to these molecules in a manner that is competitive with immune-stimulatory DNA (Kaminski *et al*, 2013; Steinhagen *et al*, 2018). However, whether pharmacologically antagonizing dsDNA-sensing cGAS via A151 could govern pyroptosis and the overall neuroinflammation in the context of brain ischemia still remains poorly defined.

# Results

## Upregulation of double-strand DNA (dsDNA) and dsDNA sensors triggered by brain ischemia

dsDNA is a potent DAMP in sterile inflammation following stroke. To detect the expression of this specific DAMP molecule in the ischemic brain, we examined dsDNA levels using immunofluorescent staining, although dsDNA staining of the sham brain was limited and distributed predominantly as ring-like nuclear contours; after the induction of MCAO, the intensities of dsDNA increased with a nucleoplasmic relocation in the penumbra; disintegration of the nucleus was also detected at time points of 6 and 24 h, as well as 3 days after MCAO (Fig 1A). Moreover, following stroke, the majority of dsDNA stainings appeared to be nuclear, with close spatial co-localization with DAPI and 53BP1, while cytoplasmic dsDNA could also be detected in the ischemic penumbra (Fig 1B). Since astrocytes and microglia are the main immune cells that quickly respond following ischemia and participate in the neuroinflammation, next, we performed double immunofluorescent analysis of dsDNA with cell type-specific markers for microglia (Iba1) and astrocytes (GFAP). Cytosolic dsDNA immunofluorescent signals could be detected in microglial Iba-1 and astrocytic GFAP (Fig 1B).

cGAS is a key cytosolic dsDNA sensor. In parallel with the heightened deposition of dsDNA during ischemia, we identified abundant expression of cGAS and STING in the ipsilateral hemisphere compared with the contralateral hemisphere on day 3 following stroke (Fig 1C and D). Immunohistochemical analysis also showed an elevation of cGAS in ischemic mouse brains, and the upregulated cGAS primarily located in the cytoplasm of cells (Fig 1E and F). Furthermore, cGAS was mainly expressed in 53BP1-positive cells, indicating that cGAS was induced where DNA double-strand breaks caused by ischemia (Fig 1G). Taken together, these data suggest that increased release of self-derived dsDNA within the cytosol may trigger the activation of cGAS-STING pathway during brain ischemia.

## Inhibition of cGAS signaling, AIM2 inflammasome, and pyroptosis by A151 after MCAO

Given the striking changes in cGAS expression of MCAO mice, modulation of cGAS may be a logical experimental approach to confer protection from the effects of MCAO. The effect of A151 on cGAS signaling was examined in brain tissues of MCAO mice. Transcript levels in brain tissues were evaluated on day 3 after MCAO. qRT–PCR revealed that cGAS and STING were highly induced in the brain of vehicle-treated MCAO animals, whereas A151 treatment abrogated the induction of both cGAS and STING (Fig 2A). Western blot analysis further confirmed that the upregulation of cGAS, STING, along with the expression of downstream NF-κB, was significantly attenuated by the administration of A151 (Fig 2B and C).

AIM2 is another dominant player of sterile inflammation in response to cytoplasmic dsDNA. It is worth noting that in addition to directly triggering by dsDNA, AIM2 can also be activated and amplified by an initiate signal type I interferón that is stimulated by cGAS signaling (Martinon *et al*, 2009; Lugrin & Martinon, 2018). Moreover, another important target of inflammasome activation is to drive pyroptosis ("fiery death"), a type of caspase-1-dependent inflammatory programmed cell death (Lamkanfi & Dixit, 2014). However, it is currently unclear whether pyroptosis occurs within brain cells following stroke and whether A151 treatment could influence AIM2 activation and pyroptosis in ischemic brain. To clarify these issues, transcript levels in brain tissues were measured on day 3 after MCAO. qRT–PCR revealed that AIM2, ASC, caspase-1, GSDMD, IL-1β, and IL-18 were highly induced in the brains of vehicle-treated MCAO animals, whereas A151 treatment inhibited the induction of these genes (Fig 2D). We also observed that administration of A151 dampened the expression of AIM2 inflammasome-related molecules (AIM2/caspase-1/ASC) at early time point of 24 h after MCAO and that A151 reduced levels of IL-1β, and several downstream inflammatory cytokines/chemokines (IL-6, MCP-1) at early time points of 6 and 24 h (Appendix Fig S1), suggesting the anti-inflammatory effect of A151 treatment initiated after the onset of cerebral ischemia. Additionally, pronounced increase of protein levels of AIM2 inflammasome components (AIM2/ASC/caspase-1), IL-1β, and pyroptosis effector GSDMD was also increased in ischemic brain compared with sham-operated group, whereas markedly repressed in MCAO mice treated with A151 (Fig 2E and F). These results suggest that AIM2 inflammasome is activated and ischemic mouse brains undergo pyroptosis after stroke, leading to the processing of pro-inflammatory factors such as IL-1β and cell death, whereas A151 could effectively suppress AIM2 activation and pyroptosis.

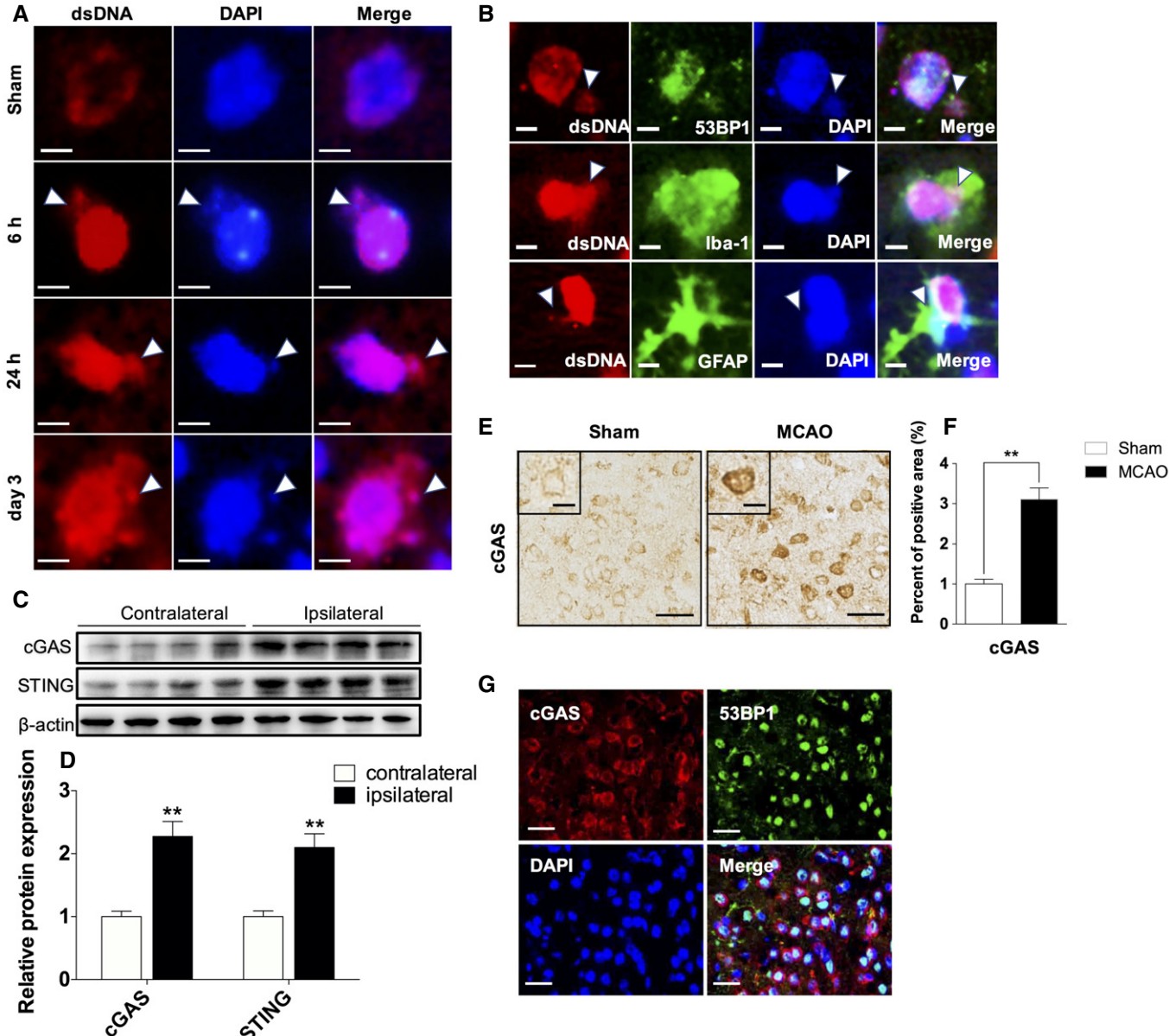

**Figure 1.  Accumulation of dsDNA and upregulation of cGAS after brain ischemia.**

A   Representative single immunofluorescent staining for dsDNA at 6, 24 h, and 3 days of reperfusion after MCAO in the penumbra as well as in the sham brain, arrowheads indicate disintegration of the nucleus. Scale bars, 5 μm.

B   Representative images of double immunofluorescent staining for 53BP-1 and dsDNA (upper panel), Iba-1 and dsDNA (middle panel), and for GFAP and dsDNA (lower panel), arrowheads indicate cytoplasmic dsDNA. Scale bars, 5 μm.

C   Western blot analysis of cGAS and STING using lysates prepared from the indicated tissues of mouse brain following MCAO.

D   Quantitative analysis for Western blot analysis. $n = 6$ mice per group. **$P < 0.01$, two-tailed unpaired Student's $t$-test.

E   Immunohistochemical staining for cGAS at 3 days of reperfusion after tMCAO in peri-ischemic area as well as in the corresponding regions of sham control brains. Scale bars, 50, 20 μm in insets.

F   Quantitative analysis of cGAS immunohistochemical staining. $n = 6$ mice per group. **$P < 0.01$, two-tailed unpaired Student's $t$-test.

G   Representative double immunofluorescent stainings for cGAS (red) and 53BP1 (green). $n = 6$ mice per group. Scale bars, 50 μm.

Data information: Data are presented as mean ± SEM. $P$-values are reported in Appendix Table S2.

## A151 prevents microglial pyroptosis after MCAO

Pyroptosis, an inflammatory programmed cell death, was recently shown to be mediated by GSDMD, which could subsequently amplify the inflammation through the concomitant release of neuro-toxic and inflammatory mediators. We then studied expression of related molecules using immunohistochemical staining. Immunohis-tochemistry analysis indicated that caspase-1, IL-1β, and GSDMD

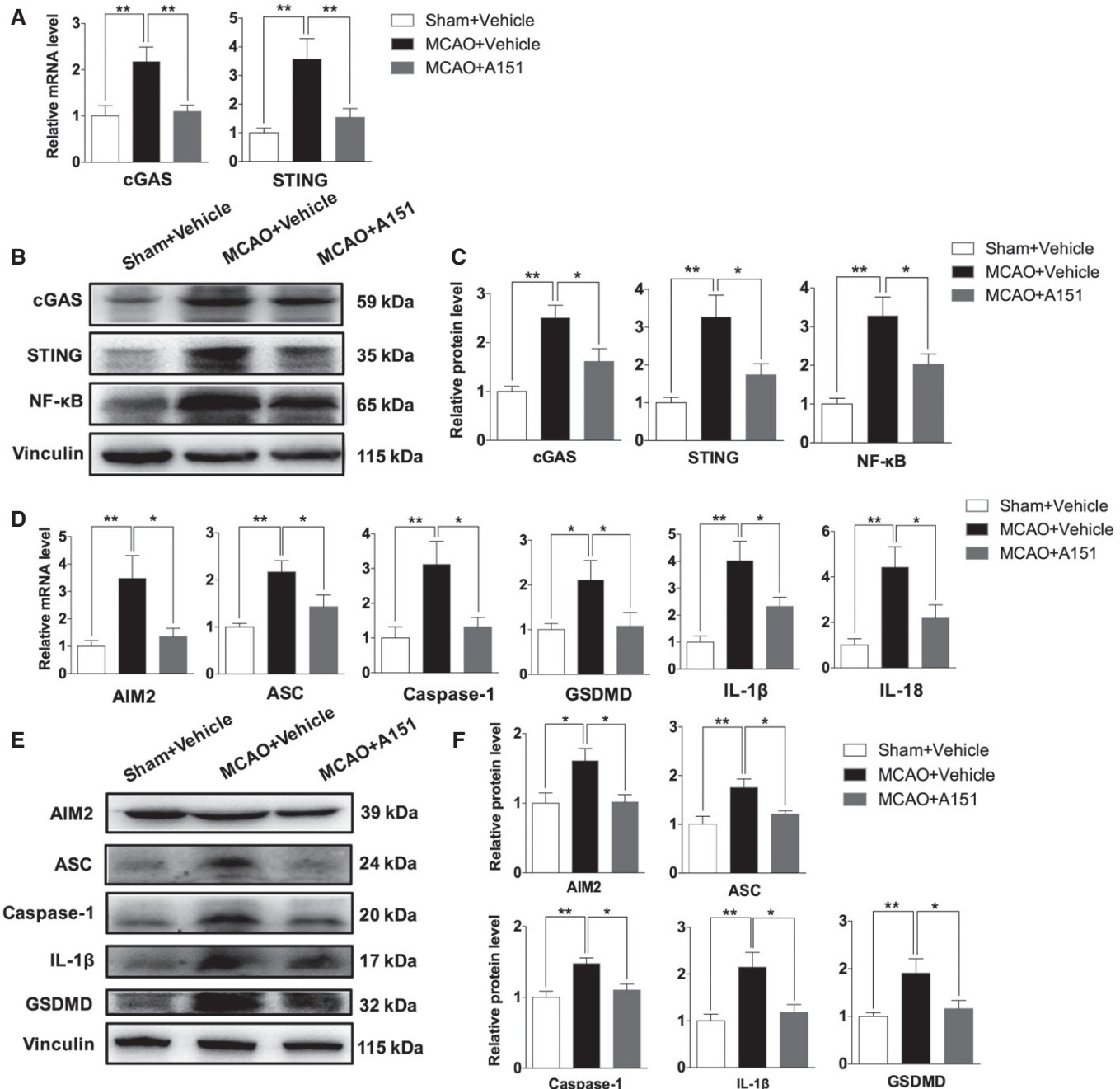

**Figure 2. A151 inhibits MCAO-induced expression of cGAS, and AIM2 inflammasome- and pyroptosis-associated molecules.**

Mice received daily intraperitoneal (IP) injections of A151 (300 μg) or an equal volume of vehicle for three consecutive days after MCAO induction.

A   The mRNA expression levels of the cGAS and STING after MCAO were determined using qRT–PCR. *n* = 6 mice per group. **P < 0.01, one-way ANOVA followed by Tukey post hoc test.

B   The expression levels of cGAS, STING, and NF-κB proteins in the brains of indicated groups were measured using Western blot analysis.

C   Bar graph shows the relative protein expression levels of cGAS, STING, and NF-κB in the brains of indicated groups. *n* = 6 mice per group. *P < 0.05, **P < 0.01, one-way ANOVA followed by Tukey post hoc test.

D   The mRNA expression levels of the AIM2 inflammasome components (AIM2, ASC, and caspase-1), effector of pyroptosis GSDMD, IL-1β, and IL-18 after MCAO were determined using qRT–PCR. *n* = 6 mice per group. *P < 0.05, **P < 0.01, one-way ANOVA followed by Tukey post hoc test.

E   The expression of AIM2 inflammasome components, GSDMD, and IL-1β in the brains of indicated groups was measured using Western blot analysis.

F   Bar graph shows the relative protein expression levels of AIM2 inflammasome components, GSDMD, and IL-1β in the brains of indicated groups. *n* = 6 mice per group. *P < 0.05, **P < 0.01, one-way ANOVA followed by Tukey post hoc test.

Data information: Data are presented as mean ± SEM. *P*-values are reported in Appendix Table S2.

expression were significantly induced in MCAO+vehicle mice (Fig 3A and B). However, the levels of these molecules were reduced with A151 treatment (Fig 3A and B). Notably, cells displayed GSDMD immunoreactivity at the plasma membrane (Fig 3A, Inset), which is consistent with pyroptosis.

Furthermore, cell type-specific analysis on GSDMD expression profiles in the ischemic penumbra showed that GSDMD

immunofluorescent signals were apparently colocalized with microglial Iba1, a few merged signals were observed in the NeuN presenting neurons, while GFAP-labeled astrocytes did not co-express GSDMD in the brain of MCAO mice (Fig 3C). These data indicate that microglia are the predominant cell subset in the central nervous system (CNS) that undergo pyroptosis in MCAO. In addition, as shown in Appendix Fig S2, we further evaluated the localization of

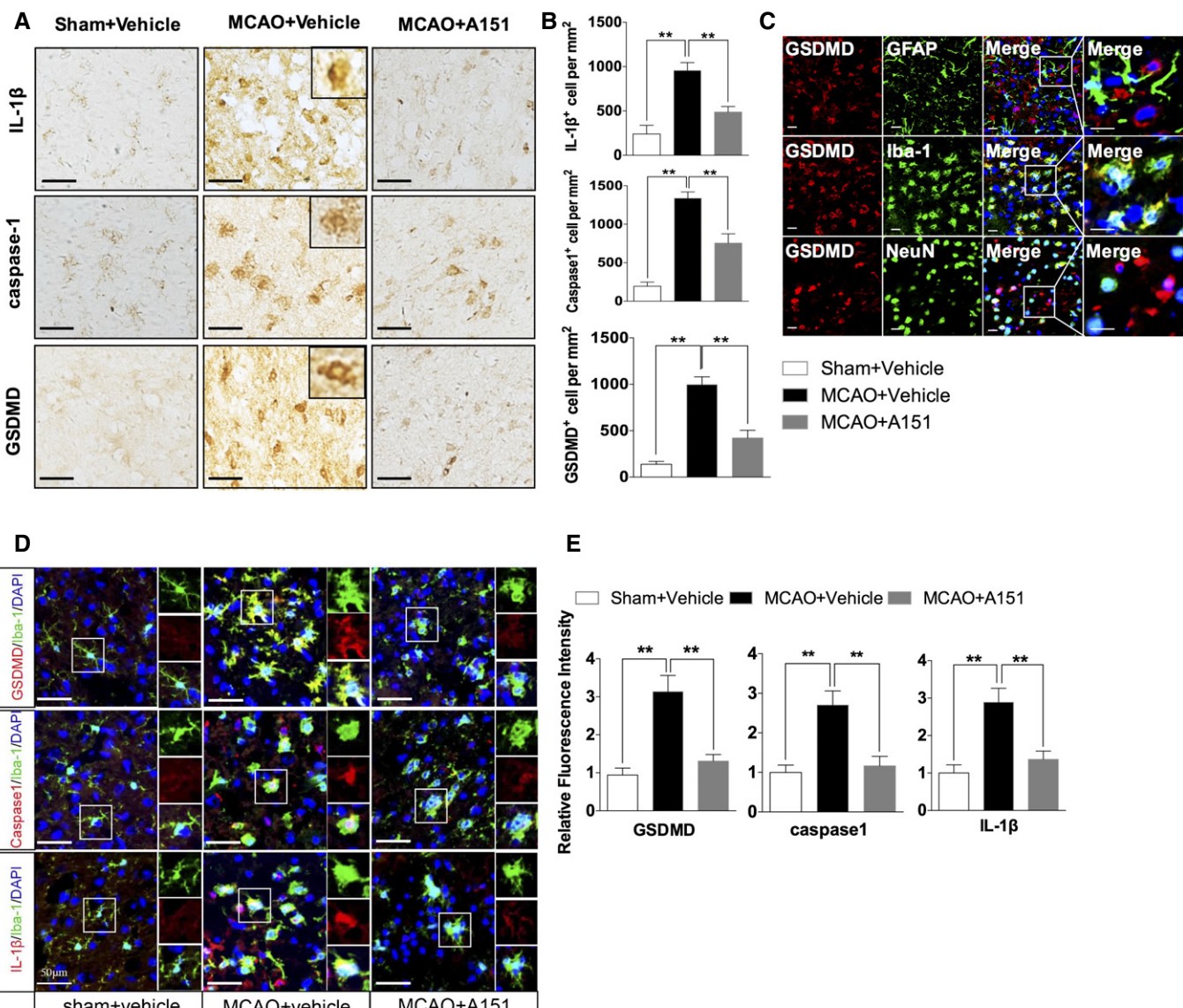

**Figure 3. A151 suppresses microglial pyroptosis after focal ischemic stroke.**

A   Representative images of immunohistochemical staining of IL-1β, caspase-1, and GSDMD in the brains of indicated group. Scale bars, 50 μm.

B   Quantitative analysis for immunohistochemical staining. n = 6 mice per group. **P < 0.01, one-way ANOVA followed by Tukey post hoc test.

C   Representative images of double immunofluorescent staining for GFAP and GSDMD, for Iba-1 and GSDMD, and for NeuN and GSDMD in post-ischemic brains after tMCAO. Scale bars, 20 μm.

D   Representative images of double immunofluorescent staining for Iba-1 and GSDMD, for Iba-1 and caspase-1, and for Iba-1 and IL-1β of sham+vehicle, MCAO+vehicle, and MCAO+A151 groups. Scale bars, 50 μm.

E   Bar graph shows the relative immunofluorescence intensity of GSDMD, caspase-1, and IL-1β in the brains of indicated groups. n = 6 mice per group. **P < 0.01, one-way ANOVA followed by Tukey post hoc test. DAPI staining: nuclei.

Data information: Data are presented as mean ± SEM. P-values are reported in Appendix Table S2.

GSDMD (red) compared to microglia marker Iba-1 (green) in the ischemic brain and found that the Iba-1 expression diminished with increasing GSDMD expression. Healthy microglia that are GSDMD⁻ exhibited the highest Iba-1 expression.

Additionally, immunofluorescent staining of GSDMD, caspase-1, and IL-1β with Iba-1 generated another line of evidence for enhanced microglial pyroptosis following stroke, whereas A151 treatment significantly suppressed such an increase (Fig 3D and E). These results suggest that A151 can prevent microglial pyroptosis in the context of ischemic conditions.

### A151 attenuates poly (dA:dT)-induced cGAS activation and caspase-1-induced pyroptosis *in vitro*

To further investigate dsDNA-sensing cGAS activation and pyroptosis in microglia with a more dsDNA-relevant stimulus, comparable analyses were performed *in vitro* by transfection of primary microglia with poly (dA:dT), a synthetic analogue of dsDNA, in combination with A151. Consistent with the *in vivo* results, the expression levels of cGAS axis and pyroptosis-associated proteins (caspase-1, IL-1β, and GSDMD) were elevated in both lysates (Fig 4A and B) and supernatants (Fig 4C and D) of LPS-primed primary microglia stimulated with poly (dA:dT) compared with the blank control group, whereas the elevation of these proteins was suppressed by A151. MTS assay and LDH analysis were performed to clarify the microglial loss *in vitro* studies. Primary microglia were treated with A151 or PBS. MTS values were normalized to PBS-treated controls and showed no difference between the two groups (Appendix Fig S3A). Lactate dehydrogenase (LDH) activity was evaluated in cell culture supernatants, poly(dA:dT) exposure increased LDH release from microglia, while A151 treatment reduced its release (Appendix Fig S3B). Immunofluorescence analysis in BV2 cells further revealed the upregulation of cGAS, IL-1β, caspase-1, and GSDMD and nuclear translocation of NF-κB induced by LPS/poly (dA:dT) stimulation (Fig 4E and F), while this induction was attenuated by A151. These results suggest that pharmacological inhibition of dsDNA-sensing cGAS and AIM2 could ameliorate dsDNA-induced pyroptosis.

### A151 suppresses neutrophil infiltration and production of microglia pro-inflammatory factors in MCAO

Next, we sought to determine the impact of A151 on brain's postischemic inflammation is mediated by modulating the activation of local resident cells (microglia) or by recruitment of peripheral immune cells, or both. We examined such cellular components in the brains of MCAO mice (Fig 5A). We found that A151 significantly reduced neutrophils (CD45^high CD11b⁺ Ly6G⁺) infiltration in the brain on day 3 after MCAO compared with those in vehicle-treated controls (Fig 5B). In addition, A151 decreased neutrophil-attracting chemoattractants CCL2 and CXCL10 (Appendix Fig S4A), which may also have a direct effect on neutrophils since several cells expressing cGAS were also positive for ly6G of neutrophils marker (Appendix Fig S4B). Interestingly, the spleens of A151-treated mice exhibited increased numbers of neutrophils but no increase of any other leukocyte subtype (Fig 5C).

In addition, microglia (CD45^int CD11b⁺) numbers decreased as did microglial expression of tumor necrosis factor-α (TNF-α) and

factor interleukin-6 (IL-6). In contrast, the microglial expression of transforming growth factor-β (TGF-β) was increased in MCAO mice receiving A151 (Fig 5D and E). An increase in IL-4 expression was also noticed, while that was not statistically significant (Fig 5E). Moreover, macrophages also co-expressed cGAS in the penumbra of MCAO animals (Appendix Fig S5A), A151 also affected the production of IL-4 and TNF-α from blood-derived macrophages (Appendix Fig S5B and C), suggesting that A151 effectively inhibited the activation of microglia/macrophage in the brain after MCAO. Double immunofluorescence staining for microglial proliferation and apoptosis as revealed by Iba-1/Ki-67 and Iba-1/TUNEL was performed in the penumbra as well as in the ischemic core in vehicle- and A151-treated mice after MCAO; A151 reduced microglial cell death and proliferation in the penumbra after stroke (Appendix Fig S6). These results indicate that A151 is capable to dampen neuroinflammation via modulating both resident microglia and peripheral inflammatory cells, and causing microglia/macrophage transformation toward an anti-inflammatory phenotype.

To further investigate the effect of A151 on neutrophils and microglia following stroke, we performed immunofluorescent staining. Immunocytochemical analysis showed that the numbers of neutrophils and microglia were reduced and the activation of microglia was attenuated in the ischemic penumbra by A151 treatment as indicated by ly6G and Iba-1 immunoreactivity (Fig 5F–I). These results support a pathogenic role of excessive dsDNA sensing on inflammatory conditions and suggest that A151 can inhibit the proliferation of microglia and the migration of both microglia and neutrophils toward damaged tissue.

### A151 protects against brain damage, improves neurodeficits, and reduces cell death after MCAO

In addition to the molecular and neuropathological findings indicating that dsDNA-sensing cGAS and AIM2 inflammasome activation, with pyroptosis induction in MCAO, were suppressed by A151 treatment, neurobehavioral assessment permitted serial analyses on the effects of the therapeutic intervention. Schematic of the experimental protocol is shown in Fig 6A. A151 treatment reduced the severity of neurobehavioral deficits assessed by mNSS, corner-turning test, foot-fault test, and adhesive removal test as early as at day 1 after MCAO and persisted to day 14 after MCAO (Fig 6B), suggesting that inhibition of cytoplasm dsDNA-sensing cGAS A151 can provide long-term benefit after MCAO. Lesion volume was measured by TTC after MCAO. The assay showed that A151-treated mice had significantly reduced lesion volume and brain edema on day 3 after MCAO, when compared with the vehicle group (Fig 6C and D). In addition, therapeutically inhibiting cGAS via A151 or cGAS inactivation by genetically deleting also exhibited significantly reduced infarcts at day 1 after MCAO, suggesting cGAS inactivation may reduce volumes of cerebral infarct in early stages of ischemic injury development (Appendix Fig S7A–D). In an effort to determine whether A151 operates in older mouse brain, we also compared stroke severity in older vehicle- and A151-treated mice. We noticed smaller cerebral infarcts and improved neurodeficits in A151-treated older mice (Appendix Fig S8A–C). We then used a propidium iodide (PI)–Annexin V kit to test for apoptosis cells on day 3 after tMCAO. A151 reduced numbers of Annexin V and PI expressing cells in the brain (Fig 6E and F). In addition, quantitation of TUNEL⁺ cells in

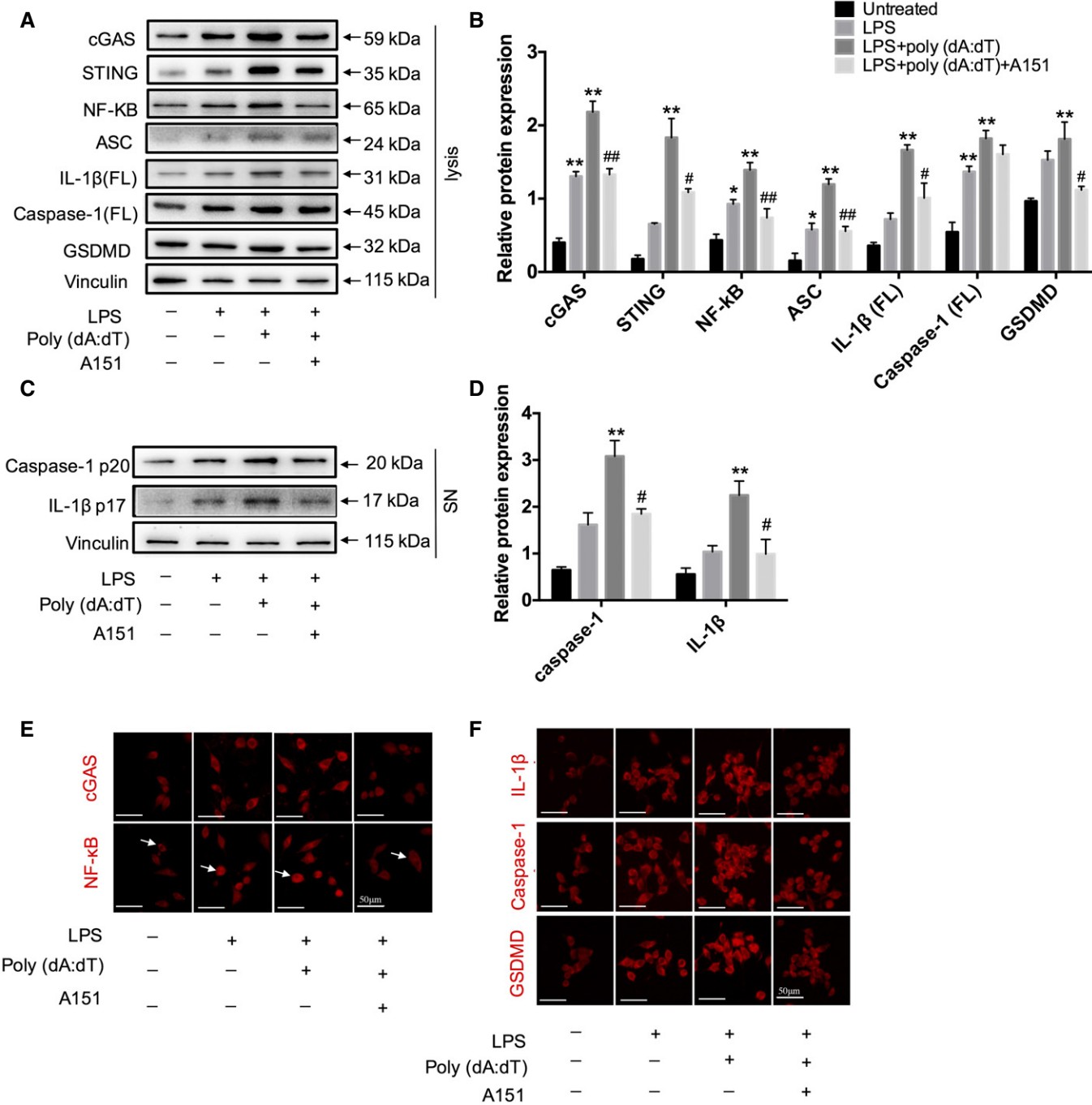

**Figure 4. A151 alleviates cGAS activity and expression of pyroptosis-associated proteins in vitro.**

A  Expression of cGAS, STING, NF-κB, ASC, IL-1β (FL), caspase-1 (FL), and GSDMD in primary microglia lysates was detected using Western blot.

B  Quantitative analysis for Western blot analysis of cGAS, STING, NF-κB, ASC, IL-1β (FL), caspase-1, and GSDMD. $n = 3$ per group. P-values are reported in Appendix Table S3, one-way ANOVA followed by Tukey post hoc test. $*P < 0.05$, $**P < 0.01$ compared with the untreated group, $^{#}P < 0.05$, $^{##}P < 0.01$ compared with the LPS+poly(dA:dT) group.

C  The protein levels of caspase-1 p20 and IL-1β (p17) in the supernatant of cultured primary microglia were determined by Western blot.

D  Quantitative analysis for Western blot analysis of caspase-1 and IL-1β. $n = 3$ per group. P-values are reported in Appendix Table S3, one-way ANOVA followed by Tukey post hoc test. $**P < 0.01$ compared with the untreated group, $^{#}P < 0.05$ compared with the LPS+poly(dA:dT) group.

E, F  Immunofluorescence was conducted to detect the expression levels of cGAS, NF-κB (arrows indicate NF-κB translocation), and pyroptosis-associated molecules (GSDMD, caspase-1, and IL-1β) in BV2 cells. $n = 3$ in each group. Scale bars, 50 μm.

Data information: Data are presented as mean ± SEM. P-values are reported in Appendix Table S2.

the penumbra area further indicated a reduction in cell apoptosis after A151 treatment (Fig 6G and H). Taken together, these data indicate that inhibition of dsDNA-sensing cGAS and AIM2 by A151 is sufficient to protect mice from ischemic insults via prompting neurological functional recovery and decreasing cell death.

## Inactivation of microglial dsDNA-sensing cGAS diminishes the protective effects of A151

Next, we determined the role of cGAS in the protective effects of A151 on ischemic brain injury. Cell type-specific analysis revealed

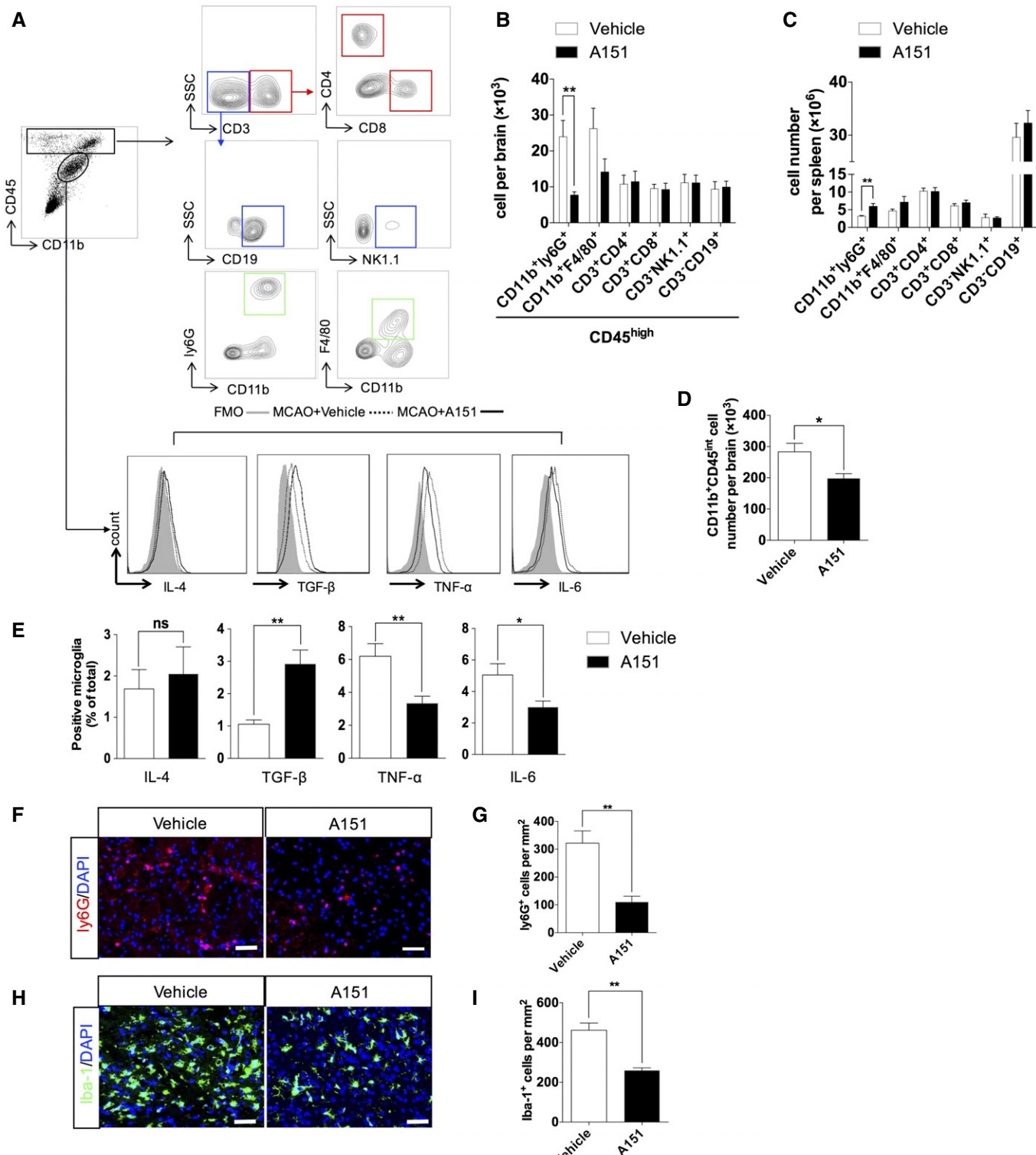

Figure 5.

**Figure 5. A151 attenuates neutrophil infiltration and production of pro-inflammatory factors in microglia after tMCAO.**

A  Gating strategy of brain-infiltrating immune cells including CD4$^+$ T cells (CD3$^+$ CD4$^+$), CD8$^+$ T cells (CD3$^+$CD8$^+$), B cells (CD3$^-$CD19$^+$), NK cells (CD3$^-$NK1.1$^+$), monocyte/macrophages (CD11b$^+$CD45$^{high}$F4/80$^+$), neutrophils (CD11b$^+$CD45$^{high}$Ly6G$^+$), and microglia (CD11b$^+$CD45$^{int}$) and their expression of IL (interleukin)-6, TNF-α (tumor necrosis factor-α), TGF-β (transforming growth factor-β), and IL-4. FMO, fluorescence minus one.

B  Quantitative analysis of the counts of brain-infiltrating leukocytes in the brains of MCAO mice receiving indicated treatment. $n = 6$ mice per group. \*\*$P < 0.01$, two-tailed unpaired Student's $t$-test.

C  Bar graph shows counts of leukocyte subtype in the spleens of MCAO mice receiving indicated treatment. $n = 6$ mice per group. \*\*$P < 0.01$, two-tailed unpaired Student's $t$-test.

D  Quantitative analysis of the microglia in the brains of MCAO mice receiving indicated treatment. $n = 6$ mice per group. \*$P < 0.05$, two-tailed unpaired Student's $t$-test.

E  Bar graph shows percentages of microglia expressing IL-6, TNF-α, TGF-β, and IL-10 in the brains of MCAO mice receiving indicated treatment. $n = 6$ mice per group. \*$P < 0.05$, \*\*$P < 0.01$, two-tailed unpaired Student's $t$-test.

F  Representative images of immunofluorescent staining for ly6G in peri-infarct areas of MCAO mice receiving indicated treatment. Scale bar, 50 μm.

G  Bar graph shows the number of ly6G$^+$ neutrophils. $n = 6$ mice per group. \*\*$P < 0.01$, two-tailed unpaired Student's $t$-test.

H  Representative images of immunofluorescent staining for Iba-1 in peri-infarct areas of MCAO mice receiving indicated treatment. Scale bar, 50 μm.

I  Bar graph shows the number of Iba-1$^+$ microglia. $n = 6$ mice per group. \*\*$P < 0.01$, two-tailed unpaired Student's $t$-test.

Data information: Data are presented as mean ± SEM. $P$-values are reported in Appendix Table S2.

that cGAS was predominantly and markedly increased in microglia after tMCAO (Fig 7A). We speculated that the genetically depletion of microglial cGAS may protect mice against ischemic injury and abolish or reduce the protective effect of A151 on ischemic brain injury. We generated double transgenic mice with microglia-specific inactivation of cGAS (knockout [KO]; cGAS-KO) by crossing conditional cGAS knockout mice carrying loxP-flanked cGAS alleles (Fig 7B) with mice expressing CX3CR1$^{CreER}$ (Fig 7C). The genotyping results are shown in Fig 7D. In addition, cGAS expression in microglia and splenic monocytes was assessed from cGAS-KO mice and littermate control (WT) mice using Western blot, and found that cGAS expression in cGAS-KO mice was downregulated in the microglia, but did not differ in splenic monocytes (Appendix Fig S9A). Double immunofluorescent staining of cGAS/Iba-1 further demonstrated the elimination of cGAS immunostaining from microglia (Appendix Fig S9B). cGAS-KO and WT mice were subjected to transient MCAO for 1 h followed by 24 h and 72 of reperfusion (Fig 7E). In comparison with the WT mice, cGAS-KO mice exhibited reduced infarct volume (Fig 7F and G) and significantly improved neurological deficits (Fig 7H). These results suggest that silencing microglial cGAS attenuates brain ischemic damage. In addition, of interest, when cGAS-KO mice were treated with vehicle or A151 after MCAO, these two groups did not differ in infarct volume and mNSS score, suggesting that the benefit of A151 was abolished in cGAS-KO mice after MCAO (Fig 7F and H), suggesting that cGAS mediates the beneficial effect of A151.

## Discussion

In the present study, we uncovered an important role for the cytosolic dsDNA-sensing cGAS in sterile inflammation after ischemic injury, pharmacological interventions aimed at antagonizing dsDNA cGAS via their inhibitor A151 attenuated the overall neuroinflammatory response by preventing microglial pyroptosis, reducing microglia activation and neutrophils infiltration, and inhibiting the release of both inflammasome-dependent cytokines (IL-1β and IL-18), as well as non-inflammasome-dependent cytokines (e.g., TNFα, IL-6). Furthermore, A151 attenuated ischemic brain injury, improved neurobehavioral performance, and reduced cell death after MCAO. Moreover, the protective effects of A151 were

abolished in cGAS-CX3CR1$^{CreER}$ mice that subjected to specific deletion of microglial cGAS. Together, these findings indicate that inhibition of dsDNA-sensing pathways may hold a potential therapeutic perspective for reducing brain ischemic injury in the clinical setting (Fig 8).

cGAS signaling is a recently described innate immune pathway of cytosolic DNA sensing (Cai et al, 2014); in the present study, we observed that cGAS is robustly activated in response to the specific DAMPs dsDNA caused by the ischemic cell death, triggering downstream pro-inflammatory events mediated by the STING cascade (Fig 1). Studies with intraperitoneal injection of ODNs of specific sequences have shown to elicit protective role through its anti-inflammatory properties in the brain of experimental mice stroke model and Alzheimer's disease (Scholtzova et al, 2009; Lu et al, 2014), indicating that the synthetic oligonucleotides could get into the brain parenchyma through the disruption of blood–brain barrier and play a role in the central nervous system. This study further confirmed the ability of A151 to effectively inhibit the activation of cGAS (Fig 2), suggesting that systemic delivery of A151 could be absorbed from the blood into the damaged brain parenchyma, and provided novel evidence that inhibition of this pathway offers the benefit of reducing MCAO injury and improving outcome (Fig 6). Type I IFN stimulated by cGAS-STING signaling has been shown to provide inflammasome-priming signals for AIM2 (Martinon et al, 2009; Labzin et al, 2016; Liu et al, 2017a). Mechanistic studies indicated that AIM2 inflammasome could contribute to brain damage and neuroinflammation after ischemic stroke (Lammerding et al, 2016). In advancing previous findings, our study for the first time provides efficacy data depicting pharmacological inhibition of AIM2 inflammasome directly or indirectly in the setting of MCAO in vivo (Fig 2). In line with the observations made in AIM2-deficient mice (Denes et al, 2015), when subjected to MCAO, pharmacological inhibition of AIM2 via A151 significantly reduced infarct volumes and dampened inflammatory response (Figs 2 and 6). Moreover, activation of AIM2 has shown to facilitate caspase-1 activation and mediate caspase-1-dependent cell death termed pyroptosis in response to endogenous dsDNA breaks (Fernandes-Alnemri et al, 2009; Hornung et al, 2009). GSDMD has recently been characterized as an effector of pyroptosis, which aggregates at the plasma membrane forming cytotoxic pores and causing a passive release of cellular contents and

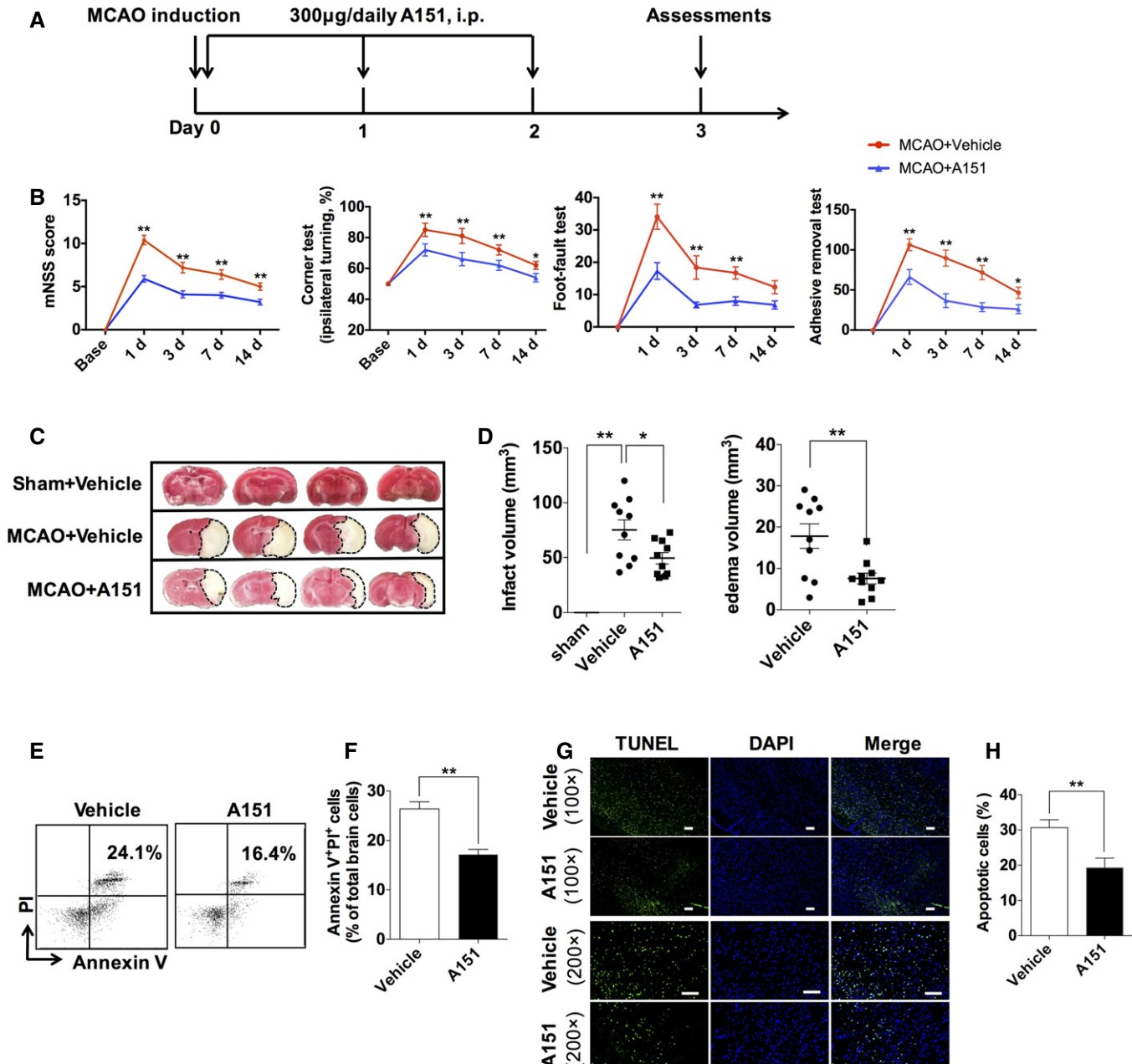

**Figure 6. A151 protects against brain damage, improves neurodeficits, and reduces cell death after MCAO.**

A Flowchart illustrates A151 administration and experimental design. Mice received daily intraperitoneal (IP) injections of A151 (300 µg) or an equal volume of phosphate-buffered saline (PBS) vehicle for 3 consecutive days starting immediately after reperfusion.

B Neurological tests were performed to evaluate the motor, sensory, and balance functions in mice receiving A151, or vehicle at days 1, 3, 7, and 14 after MCAO. $n = 10$ mice per group. *$P < 0.05$, **$P < 0.01$, two-way ANOVA with Bonferroni post hoc test.

C Representative TTC staining of brain sections from MCAO mice on day 3 after MCAO, the infarct area is shown in white.

D Bar graph shows percentages of infarct volume of indicated groups. $n = 10$ mice per group. *$P < 0.05$, **$P < 0.01$, two-tailed unpaired Student's *t*-test.

E, F Flow cytometry plots (E) and summarized results (F) show percentages of Annexin V$^+$ PI$^+$ cells in the brains of MCAO mice receiving A151 or vehicle. $n = 6$ mice per group. **$P < 0.01$, two-tailed unpaired Student's *t*-test.

G Representative histology images for TUNEL staining. Scale bars, 100 µm.

H Bar graph shows percentages of TUNEL-positive cells. $n = 6$ mice per group. **$P < 0.01$, two-tailed unpaired Student's *t*-test.

Data information: Data indicate mean ± SEM. *P*-values are reported in Appendix Table S2.

inflammasome-associated cytokines (e.g., IL-1β and IL-18) (Broz & Dixit, 2016; Aglietti & Dueber, 2017; Shi *et al*, 2017), thereby exacerbating neuroinflammation and cerebral ischemic injury. Herein, this report provides strong evidence for the activation and formation of pyroptosis following stroke, the effector molecular GSDMD along with the caspase-1 immunoreactivity accumulated on the plasma membrane, forming a distinctive "ring of fire" morphology in the ischemic brain (Fig 3). We also noted that microglia were

the predominant subset with prototypic ring of GSDMD immunoreactivity, suggesting that in ischemic conditions, brain tissues undergo pyroptosis primarily within the microglia (Fig 3).

Therefore, this experimental evidence led us to hypothesize that microglial cGAS may be a pivotal regulator of inflammatory cascades upon dsDNA recognition. We generated CX3CR1^CreER mice expressing tamoxifen-inducible Cre recombinase that allow for selective genetic cre deletion of cGAS in microglia (Parkhurst *et al*,

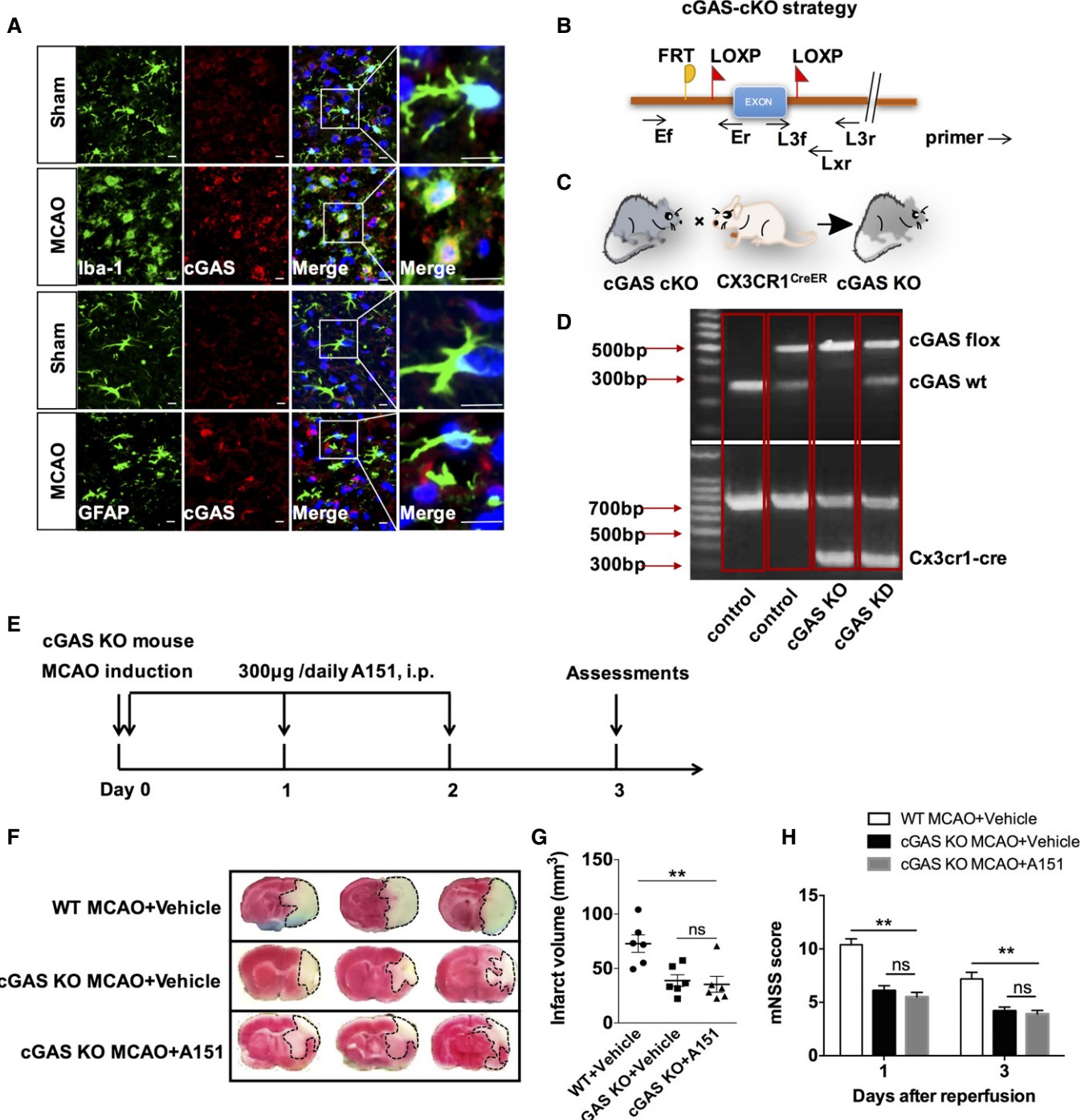

Figure 7.

**Figure 7. Microglia-specific cGAS knockout diminishes the protective effects of A151.**

A  Representative double immunofluorescent stainings for Iba-1 and cGAS, and for GFAP and cGAS. Scale bars, 10 μm.
B  Gene strategy for preparing cGAS-conditional knockout (cKO) mice.
C  Breeding scheme for CX3CR1[CreER] mice crossed with cGAS-cKO mice, in which cGAS was deleted in microglia.
D  Representative genotyping analysis for WT, cGAS-KD, and cGAS-KO mice.
E  Flowchart illustrates A151 administration and experimental design. Mice received daily i.p. injections of A151 (300 μg) or an equal volume of PBS vehicle for 3 consecutive days.
F  Representative images of TTC staining of brain sections from vehicle-treated WT MCAO mice, vehicle-treated cGAS-KO mice, and A151-treated cGAS-KO mice on day 3 after tMCAO. Three representative rostrocaudal levels of TTC staining are shown.
G  Bar graph shows percentages of infarct volume of indicated group. $n$ = 6 mice per group. **$P$ < 0.01, one-way ANOVA followed by Tukey post hoc test.
H  Bar graph shows modified neurological severity score (mNSS) in vehicle-treated WT MCAO mice, vehicle-treated cGAS-KO mice, and A151-treated cGAS-KO mice $n$ = 10 mice per group. **$P$ < 0.01, one-way ANOVA followed by Tukey post hoc test.

Data information: Data indicate mean ± SEM. $P$-values are reported in Appendix Table S2.

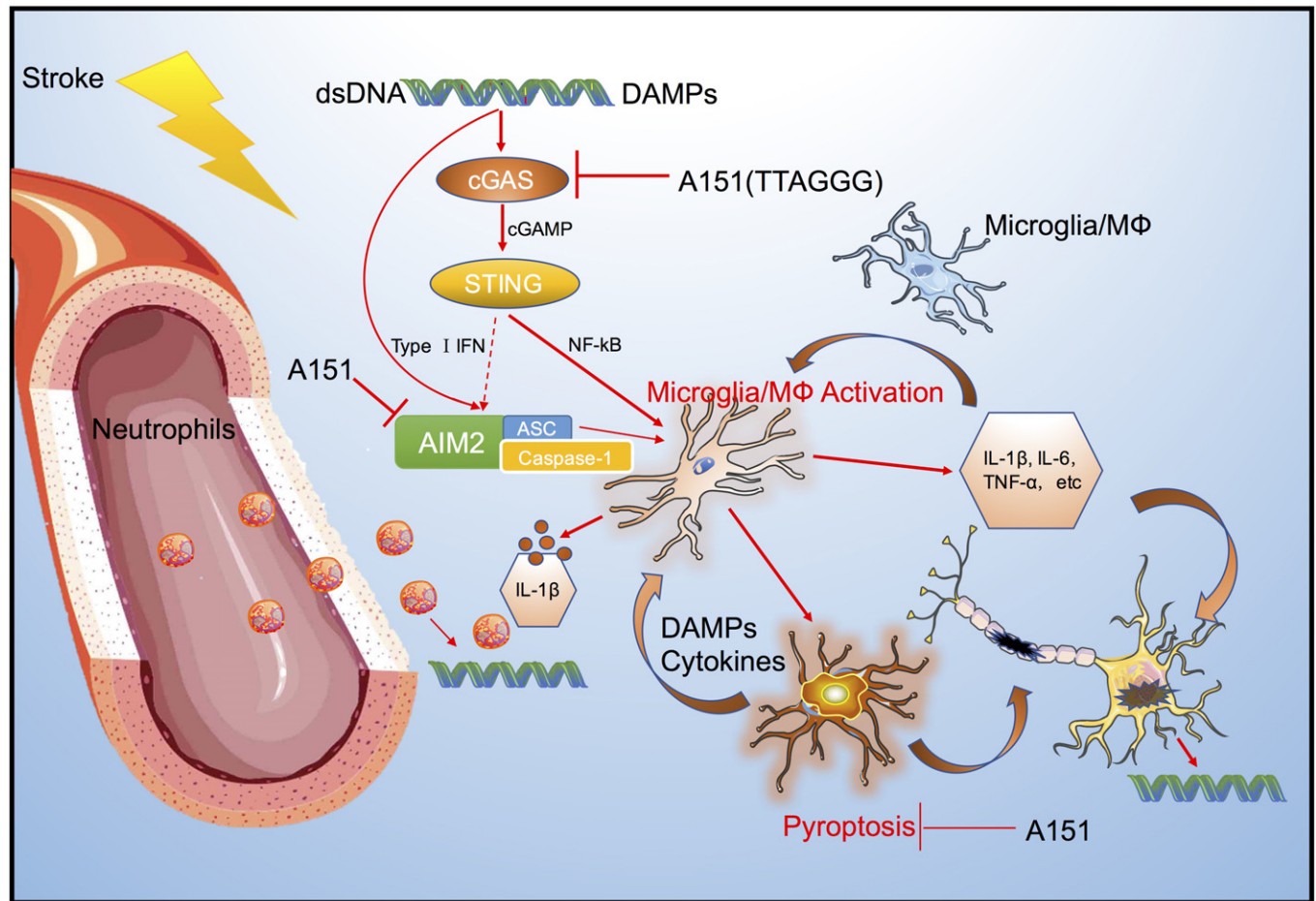

**Figure 8. A schematic diagram showing the pyroptosis and inflammatory cascades on dsDNA recognition during cerebral ischemia.**
Nuclear dsDNA released into cytoplasm by several cell death pathways during brain ischemia is a damage-associated molecular pattern, which can drive cGAS signaling and AIM2 inflammasome activation in microglia. Once activated, microglia can release both non-inflammasome-dependent cytokines (e.g., TNFα, IL-6) and inflammasome-dependent cytokines (IL-1β and IL-18). Furthermore, microglia also undergo pyroptosis that could subsequently amplify the inflammation through triggering the release of neurotoxic and inflammatory mediators. Moreover, cytokines secreted could further drive a persistent inflammatory cascade and activate infiltrating neutrophils, causing neuronal death. cGAS antagonizing synthetic oligonucleotide A151 attenuates the overall neuroinflammatory response caused by cytoplasmic dsDNA through reducing microglia activation, preventing microglial pyroptosis, and inhibiting the peripheral neutrophil infiltration. Inhibition of dsDAN-sensing cGAS and AIM2 may thus exert neuroprotection and improve stroke outcomes by interrupting this inflammatory positive feedback loop.

2013). By using this modified genetic system, we found that inactivation of microglial cGAS protected mice from ischemic injury, whereas blocked the benefit of A151 treatment (Fig 7F–H); these findings suggest that the cGAS signaling pathway is essential for A151 to exert neuroprotective activities in the context of brain ischemic injury. However, the weakness of this experiment may be

that it is not expected that reduced infarct volume of microglial cGAS-deficient mice presented to be reduced further in the experimental model; the effect of cGAS inhibition in these mice may be therefore difficult to interpret. Although the precise molecular mechanism of cGAS on neuroinflammation after ischemic stroke remains elusive and needs to be further investigated, we suspect that the interaction of downstream NF-κB with GSDMD or type I IFN with AIM2 may be involved in this process (Liu et al, 2017a,b).

Microglia are the major resident immune cells in the CNS that serve to provide the primary immune defenses; moreover, peripheral immune system also contributes to inflammatory response and cerebral infarct expansion after MCAO onset (Mabuchi et al, 2000; McColl et al, 2007; Parkhurst et al, 2013). Here, we observed that, other than decreasing local microglial activation within the ischemic penumbra, A151 also effectively attenuated neuroinflammatory response by reducing the migration of periphery neutrophils into the CNS (Fig 5). Of note, neutrophils may further exert harmful effects by subsequent release of neutrophil extracellular traps (NETs), and NETs themselves again directly allow dsDNA to be released into the microenvironment (Gallucci & Maffei, 2017), thus forming a positive feedback loop of inflammation. These findings suggest that brain resident microglia and infiltrating neutrophils may interact synergistically to coordinate dsDNA-induced inflammatory responses and culminate in the expansion of ischemic infarction. Therefore, A151 can improve the inflammatory milieu of brain and reduce inflammation-associated neuronal toxicity or death, resulting in smaller cerebral infarction and improved outcomes of stroke (Fig 7). It is worth noting that cGAS is widely distributed on a variety types of immune cells including macrophage and neutrophils (Appendix Fig S5A and Fig 4B); thus, regulation of cGAS on microglia is perhaps the main but not the exclusive mechanism responsible for the protection provided by A151. Other potential cellular targets and operating mechanisms of cGAS in MCAO await further investigations.

Robust pro-inflammatory cytokines are generally considered to be associated with a worse outcome, and cytokine-based therapies, such as inhibition of IL-1β by canakinumab (anti-IL-1β monoclonal antibody), have been approved for clinical trials (Ridker et al, 2011, 2017; Dinarello & Van der Meer, 2013). However, due to the immunomodulatory functions and pleiotropic effects of these cytokines, neutralizing, antagonizing, and blocking cytokines may come with heightened risk of unwanted side effects or unintended consequences in the clinical setting (Lambertsen et al, 2019), such as infection, and thus is a challenging process in clinical translation. Therefore, regulation of the inflammatory cascade through cGAS may take certain advantage with no perturbing of the complex and pluripotent cytokine systems compared with the use of biological inhibitors of IL-1β (Dinarello & Van der Meer, 2013). This notion is based on understanding that cGAS mediates the inflammatory response that is associated with the specific DAMP dsDNA sensing, therefore, which may enable the development of a highly targeted approach toward the cause of inflammation and may promote neuroprotection in a safe and specific manner in the clinical setting.

In summary, our findings revealed that cytosolic DNA engages multiple independent but complementary DNA-sensing signaling via cGAS and AIM2 that cooperate to maximize inflammatory responses during ischemia. This study for the first time extends the crucial role of cGAS from the field of antiviral infection (Ma & Damania, 2016), anti-tumor immunity (Wang et al, 2017), and cellular senescence (Yang et al, 2017) to brain ischemic diseases, highlighting new targets for immunotherapeutic applications. Inhibition of dsDNA-sensing cGAS is sufficient to suppress the overall neuroinflammatory response and may present a promising new therapeutic concept for ischemic stroke.

# Materials and Methods

### Ethics statement

All animal experiments were carried out according to the guidelines of the Animal Care and Use Committee of Tianjin Medical University General Hospital. All experimental procedures were performed in accordance with the recommendations of the Guide for the Care and Use of Laboratory Animals of the National Institutes of Health (NIH Publication No. 85–23, revised 1996). The report of animal experiments is in accordance with the ARRIVE guidelines (https://www.nc3rs.org.uk/arrive-guidelines).

### Animals

Eight-week-old and 12-month-old male mice were used in this study. C57BL/6 mice were purchased from Beijing Vital River Experimental Animal Technology Co., Ltd. (Beijing, China). Mice carrying cGAS-floxed alleles (cGAS$^{f/+}$) were generously provided by Hangzhou Normal University (Hangzhou, China). CX3CR1$^{CreER}$ mice were purchased from The Jackson Laboratory (#021160; Bar Harbor, ME, USA) and were maintained under specific pathogen-free conditions in the animal facility at the Chinese Academy of Medical Sciences Institute of Radiation Medicine (Tianjin, China). Mice lacking cGAS specifically in microglia were generated by crossbreeding cGAS$^{f/f}$ mice to CX3CR1$^{CreER}$ transgenic mice. Tamoxifen was given as a solution in corn oil by intraperitoneally (i.p.) for five doses of 75 mg, separated by 5 days. Four weeks after the last dose of tamoxifen, the mice were subjected to surgery. The animals were randomized into different experimental groups. To avoid any influences of sex steroids, female mice were not used in this study. Mice were allowed free access to water and food, and housed under temperature-regulated room (temperature 25–28°C and humidity 55 ± 5%) on a 12-h dark/light cycle.

### Induction of cerebral Ischemia/reperfusion model

Transient focal cerebral ischemia was induced into mice by occlusion of the left middle cerebral artery (MCA) as described previously (Longa et al, 1989). Briefly, mice were anesthetized with chloral hydrate (30 mg/kg, i.p.). During the experimental procedures, body temperature was maintained at 37 ± 0.5°C using a heat bed. The left common carotid artery was exposed through a midline cervical incision. A monofilament coated with silicone rubber (701956PK5Re; Doccol, Sharon, MA, USA) was inserted into the internal carotid artery (8–9 mm) through the common carotid artery and advanced along the internal carotid artery until it occluded the MCA. The occluding filament remained in position for the ischemic period of 60 min and was subsequently retracted to induce

reperfusion. For sham control group, animals were treated with cervical surgery but without insertion of the monofilament. The successful artery occlusion was assessed by measuring the decrease in cerebral blood flow (CBF) of the right frontoparietal cortical region by using a laser Doppler flowmeter (model ALF21; Advance, Tokyo, Japan). We defined a successful transient middle cerebral artery occlusion (tMCAO) as a > 85% decrease compared with the preischemic state CBF during ischemia and a > 80% increase of baseline CBF during reperfusion (Hochrainer *et al*, 2012). Based on the criteria above, the ratio of successful tMCAO mice was 80% of all experimental animals. The surgical incision was then closed, and the animals were allowed to recover at room temperature. All efforts were made to minimize animal suffering and reduce the number of animals used. The animals were kept at ambient temperature until sampling, with free access to water and food.

## Oligodeoxynucleotides and reagents

Synthetic oligonucleotide A151 with a phosphorothioate backbone unless otherwise specified was synthesized by TaKaRa (Dalian, China). The sequences of ODNs used in this study were 5′-T TAGGGTTAGGGTTAGGGTTAGGG-3′ (suppressive ODN: A151). Suppressive ODN A151 or its vehicle was administered i.p. at 300 μg in 0.2 ml phosphate-buffered saline (PBS) (Shirota *et al*, 2005) per mouse starting immediately after reperfusion and continuous injections once a day until the third day after surgery. The mice were randomized to treatment with A151. Poly dA:dT (Cat# tlrl-patn) was purchased from InvivoGen, and ODN TTAGGG (ODN A151) was purchased from InvivoGen (San Diego, CA).

## Assessment of neurological deficit and sensorimotor

Neurological function was evaluated in each group of mice after tMCAO by the modified neurological severity score (mNSS), corner-turning test, foot-fault test, and adhesive removal test at 1, 3, 7, and 14 days of reperfusion (Schaar *et al*, 2010; Li *et al*, 2017). The person who performed these assessment tests was blinded to the treatment received by each mouse.

## Assessment of cerebral infarction volume

2,3,5-Triphenyltetrazolium chloride (TTC) staining was used to evaluate infarct size. The mice were deeply anesthetized and decapitated, and the brains were carefully removed and manually sliced into coronal sections (1–2 mm thick) from the rostral to the caudal frontal tip with a scalpel. The slices were then incubated in 2.0% (wt/vol) TTC (Sigma, Saint Louis, MS, USA) at 37°C for 15 min, followed by overnight fixation at 4°C in 4% paraformaldehyde (PFA) (pH 7.4) (Solarbio, Beijing, China) (Li *et al*, 2017). Sections were photographed with a digital camera. The infarcted regions in each section were evaluated by using ImageJ software (National Institutes of Health). The infarct volume for each brain was calculated by the integration of infarcted areas and the distance between slices, as quantified with a computer-assisted image analyze. We also calculated the edema volume as follows: volume of the contralateral hemisphere - the volume of the ipsilateral hemisphere. To compensate for the effect of brain edema, the corrected infarct volume was calculated by using the following equation: infarct

volume × (1 − [(ipsilateral hemisphere volume − contralateral hemisphere volume)/contralateral hemisphere]) (Lu *et al*, 2013).

## Tissue preparation

The animals were deeply anesthetized after 3 days of reperfusion and were perfused with chilled PBS (pH 7.4). The brains for PCR or Western blot analysis were quickly removed, for mice that underwent MCAO treated with or without A151, the tissues were sampled from the infarct marginal zone of the left hemisphere to extract RNAs and proteins, and for the sham-operated mice, the corresponding sites in the left cerebral hemisphere were collected, then frozen in liquid nitrogen, and stored at −80°C. Brain tissues for histologic analysis were removed and immersed in 4% PFA overnight, and then were subsequently incubated in 15 and 30% (wt/vol) sucrose in PBS for 24 h at 4°C. Tissues were embedded in O.C.T. compound and frozen in liquid nitrogen, and then stored at −80°C. Successive 8-μm-thick coronal sections through the infarcted area were prepared with a freezing microtome and at −20°C and mounted on silane-coated glass slides.

## Quantitative real-time polymerase chain reaction (qPCR) analysis

Total RNAs from ischemic/reperfusion (I/R) cerebral tissues were isolated using TRIzol reagent (Invitrogen, Carlsbad, CA, USA) and quantified using a NanoDrop 2000 spectrophotometer (Thermo Scientific, Bremen, Germany). Reverse transcription was carried out with Trans-Script First-Strand cDNA Synthesis SuperMix Kit (Transgen, Beijing, China) according to the manufacturer's instructions. Subsequently, the product from reverse transcription was amplified with SYBR Green (04913850001; Roche, Basel, Switzerland) using a Bio-Rad CFX96 Detection System (Bio-Rad, Shimadzu, Japan). The primers used in quantitative PCR were shown in Table 1.

All reactions were performed in triplicate, and analysis of relative gene expression level was normalized to glyceraldehyde-3-phosphate dehydrogenase (GAPDH) using the $2^{-\Delta\Delta Ct}$ method. Melting curves were routinely performed to determine the specificity of the PCR.

## Western blot analysis

Tissue samples from the left ischemic hemisphere or culture cells were pooled and homogenized with a homogenizer or by sonication in ristocetin-induced platelet aggregation (RIPA) buffer containing protease and phosphatase inhibitors (Complete Protease Inhibitor Cocktail and PhosSTOP Phosphatase Inhibitor Cocktail; both from Roche), and then centrifuged at 12,000 *g* for 10 min at 4°C. The supernatant was collected. The total protein concentration in each sample was determined with a bicinchoninic acid (BCA) assay (QuantiPro BCA Assay Kit, Sigma-Aldrich) according to the manufacturer's instructions. Equal amount of protein samples were separated by SDS–polyacrylamide gel electrophoresis and transferred to polyvinylidene fluoride (PVDF, Immobilon-P Transfer membranes, Millipore) membranes. The membranes were blocked with 5% skimmed milk (wt/vol) in Tris-buffered saline supplemented with 0.1% Tween 20 (TBST) for 1 h at room temperature and then incubated with the corresponding primary antibodies: rabbit anti-cGAS (1:1,000; Cell Signal Technology); rabbit anti-STING (1:1,000; Cell

**Table 1. Primers for qPCR.**

| Gene | Primer, 5′–3′ | |
| --- | --- | --- |
| | Forward | Reverse |
| cGAS | AGGAAGCCCTGCTGTAACACTTCT | AGCCAGCCTTGAATAGGTAGTCCT |
| STING | CATTGGGTACTTGCGGTT | CTGAGCATGTTGTTATGTAGC |
| AIM2 | GTCACCAGTTCCTCAGTTGTG | CACCTCCATTGTCCCTGTTTTAT |
| Caspase-1 | ACAAGGCACGGGACCTATG | TCCCAGTCAGTCCTGGAAATG |
| ASC | CTTGTCAGGGGATGAACTCAAAA | GCCATACGACTCCAGATAGTAGC |
| IL-1β | GCAACTGTTCCTGAACTCAACT | ATCTTTTGGGGTCCGTCAACT |
| IL-18 | ACTTTGGCCGACTTCACTGT | GGGTTCACTGGCACTTTGAT |
| GAPDH | GCCAAGGCTGTGGGCAAGGT | TCTCCAGGCGGCACGCAGA |

Signaling Technology); rabbit anti-NF-κB p65 (1:500; Abcam); rabbit anti-AIM2 (1:1,000; Abcam); mouse anti-ASC (1:1,000; Santa Cruz Biotechnology, Dallas, TX, USA); anti-caspase-1 (1:1,000; Adipogen); rabbit anti-IL-1β (1:1,000; Cell Signaling Technology); mouse anti-GSDMD (1:1,000; Santa Cruz Biotechnology, Dallas, TX, USA), mouse anti-vinculin (1:1,000; Sigma); and mouse anti-β-actin (1:1,000; Proteintech) at 4°C overnight. The members were then washed and incubated for 1 h at room temperature with the species-appropriate horseradish peroxidase (HRP)-labeled secondary antibody (1:5,000; Transgen Biotech, Beijing, China). Protein-specific signals were detected using a Bio-Rad 721BR08844 Gel Doc Imager (Bio-Rad, Hercules, CA, USA), and the bands were quantified by densitometric analysis (ImageJ software, NIH). The relative amounts of proteins were normalized against vinculin or β-actin.

**Immunohistochemical analysis**

Immunohistochemical analysis was performed as previously described (McKenzie et al, 2018). Frozen sections (8 μm) were incubated with 0.3% hydrogen peroxide for 20 min to inactivate endogenous peroxidases. After washing in PBS, brain sections were blocked with 5% bovine serum albumin (BSA) for 1 h at room temperature. Thereafter, brain sections were incubated 4°C overnight with the following primary antibodies: mouse anti-cGAS (1:100, Santa Cruz Biotechnology); mouse anti-caspase-1 (1:100, Adipogen); goat anti-IL-1β (1:100, R&D Systems); and mouse anti-GSDMD (1:100, Santa Cruz Biotechnology). Slides were then washed in PBS and incubated with suitable biotinylated secondary antibodies (1: 500, Vector Laboratories) for 2 h at room temperature. The sections were followed by incubation with avidin–biotin–peroxidase complex (VECTASTAIN Elite ABC Kit; Vector Laboratories). The immunoreactivity was visualized with 3,3′-diaminobenzidine (DAB). A set of sections was also stained in a similar way but without the primary antibody and served as the negative control. Sections were then mounted in neutral balsam. All slides were imaged using a Nikon Coolscope digital microscope (Nikon, Tokyo, Japan).

**Immunofluorescent staining analysis**

Frozen brain sections or cells were fixed with 4% PFA for 20 min at room temperature and were permeabilized with 1% Triton X-100 (Solarbio, Beijing, China) solution for 15 min. Next, 5% BSA in PBS

with 0.1% triton was used to block non-specific binding sites at room temperature for 1 h. The slides were incubated with the following corresponding primary antibodies at 4°C overnight: mouse anti-dsDNA (1:100, Santa Cruz Biotechnology); rabbit anti-53BP1 (1:5,000, Novus Biologicals); rabbit anti-Iba1 (1:500, Wako Pure Chemical Industries, Ltd., Japan); rabbit anti-NeuN (1:500, Abcam); goat anti-GFAP (1:500, Abcam); rat anti-Ly6G (1:100, BioLegend); mouse anti-cGAS (1:100, Santa Cruz Biotechnology); mouse anti-GSDMD (1:100, Santa Cruz Biotechnology); mouse anti-caspase-1 (1:100, Adipogen); and goat anti-IL-1β (1:100, R&D Systems). The following day, the sections were washed with cold PBS; the immunoreactions were visualized using fluorescent secondary antibodies. Nuclei were costained with Fluoroshield Mounting Medium containing DAPI (104139, Abcam). All the slides were visualized and photo-documented using a confocal microscope (Olympus, Heidelberg, Germany) or a Nikon Coolscope digital microscope (Nikon, Tokyo, Japan), and quantified by ImageJ software (NIH).

**Isolation of cellular components from brains and spleens for flow cytometry analysis**

Flow cytometry was performed as previously described (Ren et al, 2018). On day 3 after MCAO and perfusion with cold PBS, the cerebral tissues obtained from mice were gently mechanically homogenized and then passed through 40-μm nylon cell strainers (Becton Dickinson, Franklin Lakes, NJ, USA) in PBS on ice. After centrifugation, the cell pellets were resuspended in 5 ml of 30% Percoll (GE Healthcare Bio Science AB, Uppsala, Sweden) and centrifuged at 700 ×g for 10 min. Cell pellets were harvested on the bottom of the tube and washed once with 5 ml 1% BSA solution for staining to block non-specific staining. Spleens were simultaneously removed under aseptic conditions, and splenocytes were harvested after lysing red blood cells. All antibodies were purchased from BioLegend (San Diego, CA, USA), unless otherwise indicated. Freshly obtained cells from either brain tissues or spleens were diluted to $1 \times 10^6$/ml and stained with antibodies directly labeled with one of the following fluorescent tags: fluorescein isothiocyanate (FITC), phycoerythrin (PE), allophycocyanin (APC), or PerCP-Cy5.5. We used the following fluorochrome-conjugated mouse reactive antibodies against CD45, CD11b, CD3, CD4, CD8, CD19, NK1.1, Ly6G, or F4/80, interleukin-6 (IL-6), IL-4, TNF-α, and TGF-β. Antibody dilutions were 1:100. Antibody stainings were performed in the dark

**The paper explained**

**Problem**

Ischemic stroke is a highly disabling neurological disease worldwide. Inflammation plays a crucial role in ischemic stroke, which expands brain damage and exists over an extended period until days thus providing more therapeutic opportunities. However, a knowledge gap exists relating to the innate inflammatory cascade triggered by cytoplasmic DNA in the context of ischemic brain.

**Results**

In this study, we report that ischemic brain injury triggers cytosolic escape of dsDNA and activates the recently described cGAS (Cyclic GMP-AMP synthase)-STING (stimulator of interferon genes) pathway. cGAS antagonist A151 abolished the cytosolic dsDNA-triggered inflammatory cascade of cGAS gene activation via modulating AIM2 inflammasome- and pyroptosis-associated proteins in mice subjected to transient middle cerebral artery occlusion. A151 effectively regulated microglia activation and decreased the infiltration of peripheral neutrophils into the injured brain following stroke, resulting in reduced infarct volume, and improved the long-term neurological outcome.

**Impact**

Collectively, these data highlight that cGAS inhibition is sufficient to govern the emergence of brain inflammation and attenuate innate sterile immune responses that promote ischemic brain injury, and identify potential new therapeutic targets for ischemic stroke.

according to their instructions; additional cell fixation and permeabilization were required for intracellular antigens staining. Cell-surface phenotype and intracellular cytokine expression were performed on a FACS Aria III flow cytometer (BD Bioscience). Data were analyzed with FlowJo software (Treestar) packages.

**Cell death analysis**

Three days after MCAO, flow cytometry was performed to evaluate cell apoptosis in the brain tissues with indicated groups using the apoptosis assay datasheet (Solarbio, Beijing, China) with a FACS Aria III (BD Bioscience, San Jose, CA, USA) according to the manufacturers' instructions. Data were analyzed with FlowJo software (Version 7.6.1, FlowJo, LLC). Terminal deoxynucleotidyl transferase dUTP nick-end labeling (TUNEL) staining was used to access the extent of cell death using a TUNEL kit (Roche, USA). Slides were observed and photographed using a Nikon Coolscope digital microscope (Nikon, Tokyo, Japan). The TUNEL-positive cells showed green nuclear staining, and all of the cells with blue nuclear DAPI staining were counted within five randomly chosen fields. The index of apoptosis was expressed as the ratio of positively stained apoptotic cells to nuclei × 100%.

**Cell culture and stimulation**

We conducted *in vitro* experiments using both primary microglia and microglia cell line, BV2 cells, which were maintained in Dulbecco's modified Eagle's medium (DMEM; Gibco, Carlsbad, CA, USA), containing 10% fetal bovine serum (FBS; Gibco), 100 U/ml penicillin/streptomycin, and incubated at 37°C with 5% $CO_2$ in a humidified atmosphere. Primary microglia isolation was performed

as previously described (Saura *et al*, 2003). Briefly, neonatal brains were obtained from the neonatal mice at 1 day of birth; the cortex sections were dissected out in HBSS containing $Ca^{2+}$ and $Mg^{2+}$ and enzymatically digested with DNase I and 0.25% trypsin–EDTA for 20 min at 37°C; after terminating the digestion, the resulting cells were filtered with a 40-μm cell strainer. The cells were suspended in growth medium (DMEM; Sigma, 10% FCS; Gibco, penicillin/streptomycin) and were plated in T75 flasks. Mixed glial cell cultures were grown for 8–10 days, and the microglia were separated from the mixed glial cell cultures through shaking the flasks at 260 rpm for 4 h at 37°C. The isolated microglia were then seeded into 6-well plates at a density of approximately $5 \times 10^5$/well and cultured for 2–3 days before harvesting. The purity of the microglia was determined by flow cytometry analysis. Briefly, mouse primary microglia or BV-2 cells were plated in 6-well plates and were primed for 3 h with 500 ng/ml lipopolysaccharide (LPS) (Sigma) before exposure to poly(dA:dT) (2 μg/ml) with or without sup ODN A151 (1 h pretreatment; 3 μM); the cells were transfected with synthetic DNA analogue poly(dA:dT) (Invivogen) using Lipofectamine 2000 (Invitrogen), according to the manufacturer's protocol, in OptiMEM (Gibco) for 6 h (Kaminski *et al*, 2013; Dick *et al*, 2016; Steinhagen *et al*, 2018). Supernatants and cell lysates were harvested and stored at −80°C before use.

**Analysis of cell viability and LDH release**

Relative cell viability and proliferation were determined by using a CellTiter 96 Aqueous One Solution Cell Proliferation Assay according to the manufacture's protocol (Promega, Madison, WI, USA). Lactate dehydrogenase (LDH) activity in primary microglia supernatants was measured as an indicator of cell death by using a cytotoxicity detection kit (Thermo Scientific). To normalize for spontaneous cell lysis, percent cytotoxicity, the percentage of cell death was calculated as follows: [(LDH sample)-(LDH negative control)]/[(LDH positive control)-(LDH negative control)] × 100 (McKenzie *et al*, 2018).

**Microglia isolation from mouse brain**

Brains were collected from PBS-perfused mice, minced, and filtered through a 70-μm cell strainer on ice, and then centrifuged at 750 *g* for 5 min. Myelin and cell debris were removed by centrifugation over a 30% Percoll Gradient (Amersham/GE Healthcare, Piscataway, NJ, USA). The cell concentration was counted and adjusted to $1 \times 10^7$ cells/ml. Subsequently, microglia were isolated by using the MojoSort™ mouse P2RY12 selection kit (BioLegend, San Diego, CA, USA), according to the manufacturer's recommended procedures. The purity of the isolated microglial cells was tested by flow cytometry.

**Monocyte isolation from the spleen**

Monocytes were isolated from mouse spleen following the given protocols of a mouse splenic monocytes isolation kit (TBD, TBD2011MPK, Tianjin, China). Briefly, the spleens were freshly isolated, gently homogenized, and then passed through a 40-μm cell sieve to give a single cell suspension. The homogenate was diluted with ice-cold buffer, the cell suspensions were centrifuged with

separation medium, and the monocytes were in the middle ivory white layer.

## Statistical analysis

We prespecified the sample size per group by power analysis using a significance level of $\alpha = 0.05$ at $\beta = 0.2$ with 80% power to detect statistical differences. SAS 9.1 software (SAS Institute Inc, Cary, NC) was used for power analysis and sample size calculations. GraphPad Prism software (GraphPad Software, Inc., La Jolla, CA, USA) was used for statistical analyses. Data were analyzed by investigators blinded to experimental treatments and grouping information. One-way ANOVA followed by appropriate *post hoc* test was used for 3 or more groups. Two-way ANOVA accompanied by a Bonferroni post hoc test was performed for multiple comparisons. Two-tailed Student's *t*-test for normal distribution or Mann–Whitney test otherwise was used for comparison between two groups. Data are expressed as the mean ± SEM. Statistical significance was accepted when $P < 0.05$.

**Expanded View** for this article is available online.

## Acknowledgements

This study was supported in part by the National Natural Science Foundation of China (81771361, 81820108014, 81571600, 81825008) and Heilongjiang Postdoctoral Science Foundation (LBH-Z19027).

## Author contributions

QL designed and performed experiments and wrote the manuscript; QL, YC, and CD conducted *in vivo* experiments and analyzed data; QL, BH, and RH conducted *in vitro* experiments and contributed to data analysis; HM analyzed the data; LW and JH conceptualized the study and supervised the experiments; and LW revised the manuscript. All authors read and approved the final manuscript.

## Conflict of interest

The authors declare that they have no conflict of interest.

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
