## [Review Process File · EMBO Molecular Medicine]

Inhibition of double-strand DNA sensing cGAS ameliorates brain injury after ischemic stroke

Qian Li, Yuze Cao, Chun Dang, Bin Han, Ranran Han, Heping Ma, Junwei Hao and Lihua Wang

Review timeline:

Submission date:	14 th June 2019
Editorial Decision:	8 th August 2019
Appeal received:	13 th August 2019
Editorial Decision:	9 th October 2019
Revision received:	7 th January 2020
Editorial Decision:	11 th February 2020
Revision received:	23 rd February 2020
Accept:	25 th February 2020

Editor: Jingyi Hou

Transaction Report:

1st Editorial Decision

8th August 2019

Thank you for submitting your work to EMBO Molecular Medicine. We have now received feedback from the three reviewers whom we asked to evaluate your manuscript. As you will see below, the reviewers raise substantial concerns about your work, which unfortunately preclude its publication in EMBO Molecular Medicine.

Overall, the reviewers acknowledge the potential interest of the study. However, they raise serious concerns with regard to the adequacy of the mouse model and cell line, the lack of assessment of long-term outcomes, insufficient evidence for microglial pyroptosis, and the lack of data demonstrating A151 brain penetration. I am therefore afraid that the reviewers are not convinced of the conclusiveness of the reported findings. Since clear and conclusive insights into a novel clinically relevant observation are key for publication in EMBO Molecular Medicine, and together with the fact that we only accept papers that receive enthusiastic support upon initial review, I am afraid that we cannot offer to consider the manuscript further.

***** Reviewer's comments *****

Referee #1 (Comments on Novelty/Model System for Author):

Only young, male mice are used and only short-term outcomes are assessed, which limit the medical impact.

Referee #1 (Remarks for Author):

The authors provide the results of a series of experiments aimed to determine the therapeutic efficacy of inhibiting cytosolic dsDNA on inflammation and outcomes after experimental cerebral

ischemia. They utilize immunofluorescence, Western blotting, PCR, in vivo and in vitro models, and transgenic mice to confirm the role of cGAS in post-stroke inflammation. Therapies targeting post-stroke inflammation represent an important potential treatment for stroke, and thus the work is considered of potentially high impact.

The activation of cGAS is attributed to microglia but blood-derived macrophages have infiltrated the area by day 3 and Iba1 cannot differentiate the cell types. Were there differences in cytokine production by flow cytometry within the macrophages as well? Only the microglia data is shown in Figure 5.

They have an interesting finding of reduced neutrophil infiltration in the A151-treated mice. What cells produce neutrophil chemokines in the brain after ischemia and were these suppressed by A151 or could there be a direct effect on A151 on neutrophils?

Several requirements for preclinical stroke modeling are missing from the methods. Were sample sizes prespecified? Were mice randomized to treatment with A151? How? They should justify the use of only young, male mice.

Only a single time point (day 3) is shown. Does the inhibition of cGAS lead to long term improvements in outcome?

Minor English grammar errors require editing.

Referee #2 (Comments on Novelty/Model System for Author):

A potentially interesting paper, but there are major conceptual issues that limit my enthusiasm in its present form. Key issues are the missing evidence for the role of microglial pyroptosis in brain injury in this model and the assessment of exclusively delayed inflammatory responses that weaken the authors claims. Please refer to the corresponding details and other points in my assessment.

Referee #2 (Remarks for Author):

In the research paper by Li et al., the authors show that the Cyclic GMP-AMP synthase (cGAS) antagonist A151 reduces brain injury and inflammation after experimental stroke in mice. They find that administration of A151 dampens the expression of cGAS, AIM2, caspase-1, gasdermin D and IL-1 β /IL-18 in the ischemic brain tissue 3 days after stroke, which parallels reduced microglial numbers, microglial cytokine production and neutrophil counts. Microglial cGAS deficiency is also associated with smaller infarct size and improved neurological outcome. Based on these data, the authors conclude that cytosolic DNA sensing via cGAS and AIM2 collectively amplify inflammatory responses during brain ischemia and inhibition of dsDNA-sensing cGAS is sufficient to attenuate inflammation and brain injury, providing novel therapeutic opportunities for ischemic stroke.

Novel discoveries in the paper include the protective role of A151 in experimental stroke and that brain injury is reduced in the absence of microglial cGAS in mice. These results are potentially interesting for the field. There are, however several major conceptual and technical issues that limit the impact of the present studies.

Specific points:

- The authors emphasize the role of dsDNA sensing cGAS in dampening inflammation as being a crucial mechanism for brain protection. To justify this conclusion, dsDNA staining is performed and microglia, astrocytes, inflammasome-related molecules and inflammatory cytokines are measured as late as 3 days after stroke. Since based on a number of studies, blockade of inflammatory mediators (including inflammasome- and IL-1-related pathways) is primarily effective in the first hours up to one day after acute brain injury when proinflammatory cascades are induced, it is not possible to draw appropriate conclusions from this data set regarding the mechanisms through which cGAS inhibition reduces brain injury. At 3 days anything measured in the brain may be secondary to smaller infarct size, while it remains unclear if inflammatory mediators are altered between 4h and

24h after stroke when the majority of the infarct is formed. In line with this, levels of IL-1 β and several downstream inflammatory cytokines/chemokines (IL-6, MCP-1, etc) are known to peak between 4h and 24h after cerebral ischemia in mice. The authors must provide comparative assessment of inflammasome activation and microglial responses at earlier time points (ideally at 4-8h and 24h post-stroke) to support their conclusions. Similarly, dsDNA staining at 3 days post-stroke may simply indicate the presence of heavily infarcted tissue and it remains unclear whether similar increases are seen at earlier time points to influence microglial inflammatory responses at the critical time window.

- The authors suggest that dsDNA triggers microglial pyroptosis, which is prevented by A151 leading to reduced brain injury. They show that microglia are GSDMD-positive and conclude that microglia are the main cells undergoing pyroptosis after MCAo. There several problems with these interpretations. First, the authors do not show microglial cell death with specific markers. Second, they show by using both flow cytometry and histology that A151 treatment is associated with less microglia in the brain 3 days after stroke (Figure 5). If microglia die and A151 reverses this, one would expect more microglia to survive. If the authors claim (although they do not prove) that microglial pyroptosis is important in disease pathophysiology in the current model, they should at least follow microglial numbers over time in both the infarct core and the peri-infarct tissues and assess cells death and proliferation simultaneously to make these claims sound. In Figure 5H, morphologically activated microglial cells are shown and these are reduced in numbers in A151-treated mice, which raises the possibility that A151 actually increases microglial cell death or reduces their proliferation/migration. Did the authors observe any microglial loss in their in vitro studies?

- Neurological assessment suggests that brain protection by A151 may be apparent already at 24h post-stroke. Would this imply that microglial pyroptosis as assessed at 3 days reperfusion is not a crucial contributor to disease pathophysiology? To overcome this controversy, 24th infarct size data are strongly required after both A151 treatment and in microglial cGAS deficient mice.

- The TTC images presented in Figures 6C and 7F are not in line with the percentage infarct volume data as shown on the graphs. In fact, 60min MCAo should result in much larger injury than 20-30% of the ipsilateral hemisphere. Please give details how exactly infarct size was measured and provide individual data points for given mice on all graphs reporting infarct volume. Has oedema correction been used? Did brain oedema differ between vehicle and A151-treated mice?

- Microglial cGAS-deficient mice have reduced infarct size and based on the data presented it is unlikely that this could be reduced any further in this experimental model. Therefore, the lack of an effect of cGAS inhibition in these mice is difficult to interpret. The authors should revise related statements in the manuscript. Elimination of cGAS immunostaining from microglia and its overall reduction in microglial cGAS KO mice should be demonstrated.

Further points:

- ds DNA staining shown in Figure 1 does not colocalize with DAPI and is not seen in the contralateral side. What type of changes in DNA structure would this antibody recognize given that the staining is specific? Also, microglia only occasionally show dsDNA staining, while several non-microglial cells do.

- Figure 1E: β -actin levels are highly variable making the interpretation of STING and cGAS western blotting results difficult.

- Figure 3A: Immunostaining against IL-1 β and caspase-1 appears non-specific with high background. Please provide appropriate controls.

- Figure 6: What type of cells show Annexin V and PI signal 3 days after MCAo? If these are neurons, and there is substantial delayed injury-related neuronal death 3 days after MCAo in these studies is there a significant increase in infarct size between 24h and 3 days post-stroke?

- Did the authors check whether 4 weeks after tamoxifen treatment was sufficient for peripheral Cx3cr1-positive monocytes/macrophages to be replaced by other cells? Splenic cGAS expression in

Cx3Cr1-positive monocytes should be shown.

- Is there any data showing brain penetration or pharmacokinetics of A151 after intraperitoneal injection?

Referee #3 (Comments on Novelty/Model System for Author):

1. In vivo model: long term outcomes are completely missing.
2. In vitro model: the use of BV2 cell line is inadequate.

Referee #3 (Remarks for Author):

This is a very interesting study that tested an interesting hypothesis regarding the role of dsDNA-induced cGAS and AIM2 pro-inflammatory pathways in a murine model of focal stroke and reperfusion. The authors also showed that administration of A151, a synthetic inhibitor for the aforementioned pathways conferred neuroprotection against ischemic injury.

While the study appears to be well designed and the ideas are interesting, there are many flaws in the data, which reduce the enthusiasm.

Main points:

1. A151 is synthetic oligonucleotides. In order to reach the targets, the systemically administrated A151 (i.p.) would have to go across the blood brain barrier, get into the brain parenchyma, and accumulate in the microglia with certain concentrations. The authors need to present evidence that A151 is indeed capable of doing all these. By the way, how did the authors determine the dose they used in the study?
2. As the authors correctly stated, the effect of microglial activation after brain ischemia is perhaps mainly the secondary brain injury. Unfortunately the stroke outcomes were assessed up to 3 days after ischemia. Long term outcomes are completely missing.
3. The quality of some key data is questionable. For instance, most Western blots were bleached, and many bands were oversaturated. The problems can be seen in Figure 1E, Figure 2B, Figure 2E, Figure 4A and Figure 4C.
4. All in vitro experiments were done using the BV2 cell line. This cell line has many problems. To the least, results from BV2 cells would need to be confirmed using primary cultures.

Relatively minor points:

1. Figure 1 C, not sure that dsDNA is cytosolic in Iba-1 cells as claimed, as the dsDNA staining appears to co-localized with DAPI. Also, some GFAP+ cells also contain dsDNA.
2. Figure 1D. Again, the majority of dsDNA staining appears to be nuclear, with close spatial co-localization with DAPI and 53BP1.
3. Figure 2. Unclear where the tissues were sampled from the stroke brains for the various assays. Not sure why A151 reduced the protein complexes of AIM2 and ASC, and caspase-1 at both mRNA and protein levels.
4. Figure 3A. Not sure if any evidence for pyroptosis. The authors claimed that the GSDMD was found at cell plasma membrane; therefore this was consistent with pyroptosis. However, this is hardly a supporting evidence for such a conclusion. The images could not tell whether it's cytosolic or cell membrane staining. No evidence was provided to indicate whether these GSDMD positive Iba-1 cells were live or dead/dying.

Preliminary response to Reviewer's comments.

***** Reviewer's comments *****

Referee #1 (Comments on Novelty/Model System for Author):

Only young, male mice are used and only short-term outcomes are assessed, which limit the medical impact.

Thank you very much for the suggestion, we will also use old mice (12 months) to assess the therapeutic efficacy of A151.

We will also assess long-term outcome until 21 days, we determine to assess the neurological deficits and sensorimotor at the time points of 1, 3, 7, 14, 21 days.

Referee #1 (Remarks for Author):

The authors provide the results of a series of experiments aimed to determine the therapeutic efficacy of inhibiting cytosolic dsDNA on inflammation and outcomes after experimental cerebral ischemia. They utilize immunofluorescence, Western blotting, PCR, in vivo and in vitro models, and transgenic mice to confirm the role of cGAS in post-stroke inflammation. Therapies targeting post-stroke inflammation represent an important potential treatment for stroke, and thus the work is considered of potentially high impact.

The activation of cGAS is attributed to microglia but blood-derived macrophages have infiltrated the area by day 3 and Iba1 cannot differentiate the cell types. Were there differences in cytokine production by flow cytometry within the macrophages as well? Only the microglia data is shown in Figure 5.

Thank you very much for the question, it is very important. We will do double immunofluorescence staining of cGAS/CCR2(blood-derived macrophages marker), or cGAS/P2Y12 (microglia specific marker) to differentiate the cell types of microglia and blood-derived macrophages.

Flow Cytometry analysis will be performed using the following fluorochrome-conjugated mouse reactive antibodies against CD45, CD11b, F4/80, interleukin-6 (IL-6), IL-4, TNF- α , and TGF- β to measure the cytokine production within the macrophages.

They have an interesting finding of reduced neutrophil infiltration in the A151-treated mice. What cells produce neutrophil chemokines in the brain after ischemia and were these suppressed by A151 or could there be a direct effect on A151 on neutrophils?

Thank you very much for the question, the recruitment of neutrophils into tissue is orchestrated by tissue-resident cells, which could release multiple chemokines or adhesion molecules [1,2,3], such as CCL2, CCL5, CXCL1, CXCL5, and CXCL10, thus we determine to perform PCR to analyze the expression level of CCL2, CCL5, CXCL1, CXCL5, and CXCL10. Double immunofluorescence staining of Ly6G/cGAS will be performed to clarify whether there is a direct effect on A151 on neutrophils.

Several requirements for preclinical stroke modeling are missing from the methods. Were sample sizes prespecified? Were mice randomized to treatment with A151? How? They should justify the use of only young, male mice.

Thank you very much for the question. Actually, previous studies have used various mice types to make models of middle cerebral artery occlusion including female mice, male mice, young mice, and old mice. Estradiol is a protective factor in the adult and aging brain against cerebral ischemia [4,5].

According to epidemiological data, incidence of ischemic stroke increases with age, which occurs frequently in middle-aged and elderly people, and is higher in men than in women. Mice at 8 weeks are middle-aged mice, thus we chose 8 weeks male mice to do the experiment, which is commonly used in the research field of ischemic stroke [6,7]. As for the old mice, it is necessary, and we will use elderly mice (12 months) to assess the role of inhibition of cGAS in supplementary experiments, since ischemic stroke is more common in the elderly.

Thank you very much for the question. Yes, sample sizes were prespecified, and the number of mice in each group is generally determined according to the type of experiment, such as neurological scoring, the variation among mice in the group is large, so the number is generally large, about 10 mice per group, while the number of mice used in quantitative index analysis such as flow cytometry is relatively small, about 4-6 mice per group, which were also based on the previous studies [6,8,9], and are approved and recognized in the field. The mice were randomized to treatment with A151.

Only a single time point (day 3) is shown. Does the inhibition of cGAS lead to long term improvements in outcome?

Thank you for the suggestion, it is very important. We will also assess long-term outcome until 21 days, we determine to assess the neurological deficit and sensorimotor to determine the role of inhibition of cGAS at the time points of 1, 3, 7, 14, 21 days.

Minor English grammar errors require editing.

Thank you for the suggestion, we will edit English grammar errors.

Referee #2 (Comments on Novelty/Model System for Author):

A potentially interesting paper, but there are major conceptual issues that limit my enthusiasm in its present form. Key issues are the missing evidence for the role of microglial pyroptosis in brain injury in this model and the assessment of exclusively delayed inflammatory responses that weaken the authors claims. Please refer to the corresponding details and other points in my assessment.

Thank you very much for the suggestions, as for the role of microglial pyroptosis in brain injury in this model, one of the defining features of inflammasome activation is the induction of a pro inflammatory programmed cell death termed pyro- (fire/fever) ptosis (falling) [10]. It has been demonstrated that the canonical inflammasome pathway aggravates brain ischemic injury through the spread of inflammation after stroke, while inhibiting the inflammasome pathway activation provides protective effects in experimental stroke models [10]. Pyroptosis is an active caspase-1 mediated pro-inflammatory lytic programmed cell death through cleavage of gasdermin D (GSDMD). The GSDMD forms membrane pores, which causes cell swelling and subsequently amplify the inflammation through the concomitant release of neurotoxic and inflammatory mediators [11], indicating that pyroptosis is a potential target for ischemic stroke intervention.

Moreover, we will assess the inflammatory responses at early time points of 6 h and 24 h.

Referee #2 (Remarks for Author):

In the research paper by Li et al., the authors show that the Cyclic GMP-AMP synthase (cGAS) antagonist A151 reduces brain injury and inflammation after experimental stroke in mice. They find that administration of A151 dampens the expression of cGAS, AIM2, caspase-1, gasdermin D and IL-1 β /IL-18 in the ischemic brain tissue 3 days after stroke, which parallels reduced microglial numbers, microglial cytokine production and neutrophil counts. Microglial cGAS deficiency is also associated with smaller infarct size and improved neurological outcome. Based on these data, the authors conclude that cytosolic DNA sensing via cGAS and AIM2 collectively amplify inflammatory responses during brain ischemia and inhibition of dsDNA-sensing cGAS is sufficient to attenuate inflammation and brain injury, providing novel therapeutic opportunities for ischemic stroke.

Novel discoveries in the paper include the protective role of A151 in experimental stroke and that brain injury is reduced in the absence of microglial cGAS in mice. These results are potentially interesting for the field. There are, however several major conceptual and technical issues that limit the impact of the present studies.

Specific points:

- The authors emphasize the role of dsDNA sensing cGAS in dampening inflammation as being a crucial mechanism for brain protection. To justify this conclusion, dsDNA staining is performed and microglia, astrocytes, inflammasome-related molecules and inflammatory cytokines are measured as late as 3 days after stroke. Since based on a number of studies, blockade of inflammatory mediators (including inflammasome- and IL-1-related pathways) is primarily effective in the first hours up to one day after acute brain injury when proinflammatory cascades are induced, it is not possible to draw appropriate conclusions from this data set regarding the mechanisms through which cGAS inhibition reduces brain injury. At 3 days anything measured in the brain may be secondary to smaller infarct size, while it remains unclear if inflammatory mediators are altered between 4h and 24h after stroke when the majority of the infarct is formed. In line with this, levels of IL-1 β and several downstream inflammatory cytokines/chemokines (IL-6, MCP-1, etc) are known to peak between 4h and 24h after cerebral ischemia in mice. The authors must provide comparative assessment of inflammasome activation and microglial responses at earlier time points (ideally at 4-8h and 24h post-stroke) to support their conclusions. Similarly, dsDNA staining at 3 days post-stroke may simply indicate the presence of heavily infarcted tissue and it remains unclear whether similar increases are seen at earlier time points to influence microglial inflammatory responses at the critical time window.

Thank you very much for this useful suggestion. We will perform PCR analysis of AIM2, ASC, caspase-1, IL-1 β , IL-6, and MCP-1 to provide comparative assessment of inflammasome activation and microglial responses in vehicle-treated and A151 treated group at earlier time points of 6 h and 24 h after stroke.

Immunofluorescence staining of dsDNA staining or western blot analysis will also be performed to clarify whether similar increases can be seen in vehicle-treated and A151 treated group at earlier time points of 6 h and 24 h after stroke to influence microglial inflammatory responses.

- The authors suggest that dsDNA triggers microglial pyroptosis, which is prevented by A151 leading to reduced brain injury. They show that microglia are GSDMD-positive and

conclude that microglia are the main cells undergoing pyroptosis after MCAo. There are several problems with these interpretations. First, the authors do not show microglial cell death with specific markers. Second, they show by using both flow cytometry and histology that A151 treatment is associated with less microglia in the brain 3 days after stroke (Figure 5). If microglia die and A151 reverses this, one would expect more microglia to survive. If the authors claim (although they do not prove) that microglial pyroptosis is important in disease pathophysiology in the current model, they should at least follow microglial numbers over time in both the infarct core and the peri-infarct tissues and assess cell death and proliferation simultaneously to make these claims sound. In Figure 5H, morphologically activated microglial cells are shown and these are reduced in numbers in A151-treated mice, which raises the possibility that A151 actually increases microglial cell death or reduces their proliferation/migration. Did the authors observe any microglial loss in their *in vitro* studies?

Thank you very much for the question, in the present study, thus we examined whether pharmacologically antagonizing dsDNA-sensing cGAS via A151 could govern pyroptosis and the overall neuroinflammation in the context of brain ischemia.

Herein, we provide evidence for the activation and formation of pyroptosis following stroke, the effector molecular GSDMD along with the caspase-1 immunoreactivity accumulated on the plasma membrane, forming a distinctive “ring of fire” morphology in the ischemic brain (Fig 3), consistent with pyroptosis observed in a previous study in EAE brain tissue sections[12]. Moreover, A151 effectively reduced the expression of cGAS, AIM2 inflammasome, and pyroptosis related molecules, including caspase-1, gasdermin D, IL-1 β , and IL-18.

As for the number of microglia, the number of microglia = microglial proliferation/migration - microglial cell death. However, this may not mean that A151 actually increases microglial cell death or reduces their proliferation/migration, the final microglia number depends on the degree of both microglial cell death and their proliferation/migration. The number of microglia $\downarrow\downarrow$ = their proliferation/migration $\downarrow\downarrow$ - microglial cell death \downarrow , which may also lead to a less number of the microglia.

We will perform apoptosis flow-cytometric analyses of microglia and proliferation of microglia as revealed by FACS analysis of Ki-67-stained cells in vehicle-treated and A151 treated mice after tMCAO to clarify these issues.

LDH analysis or placental blue staining will be performed to clarify the microglial loss in *in vitro* studies.

- Neurological assessment suggests that brain protection by A151 may be apparent already at 24 h post-stroke. Would this imply that microglial pyroptosis as assessed at 3 days reperfusion is not a crucial contributor to disease pathophysiology? To overcome this controversy, 24h infarct size data are strongly required after both A151 treatment and in microglial cGAS deficient mice.

Thank you very much for the question. Inflammation and immune responses have emerged as important elements in the onset and progression of stroke, microglia play an important role in inflammation[13]. By 3 days, microglia reach their peak number and peak inflammatory response in the peri-ischemic region, thus we chose this peak point to assess the effect of A151 on microglial pyroptosis. But this does not imply that microglial pyroptosis as assessed at 3 days reperfusion is not a crucial contributor to disease pathophysiology.

We will perform TTC staining to acquire the 24 h infarct size data after both A151 treatment and in microglial cGAS deficient mice.

-The TTC images presented in Figures 6C and 7F are not in line with the percentage infarct volume data as shown on the graphs. In fact, 60min MCAo should result in much larger injury than 20-30% of the ipsilateral hemisphere. Please give details how exactly infarct size was measured and provide individual data points for given mice on all graphs reporting infarct volume. Has oedema correction been used? Did brain oedema differ between vehicle and A151-treated mice?

Thank you very much for the suggestion, we will redo the analysis to acquire the exact infarct size. The total infarct volume for the entire brain will be calculated as the sum of the infarct volume of each section. To compensate for the effect of brain edema, the infarct volume percentage was calculated as follows: $\text{infarct area} \times (1 - [(\text{ipsilateral hemisphere area} - \text{contralateral hemisphere area}) / \text{contralateral hemisphere}])$. Infarct volumes were calculated by the integration of infarcted areas on each brain slice, as quantified with a computer-assisted image analyzer [14].

Individual data points will also be provide for given mice on all graphs reporting infarct volume.

The new formula for the infarct volume correct the edema, and we will also compare the brain edema to show whether it differs between vehicle and A151-treated mice.

- Microglial cGAS-deficient mice have reduced infarct size and based on the data presented it is unlikely that this could be reduced any further in this experimental model. Therefore, the lack of an effect of cGAS inhibition in these mice is difficult to interpret. The authors should revise related statements in the manuscript. Elimination of cGAS immunostaining from microglia and its overall reduction in microglial cGAS KO mice should be demonstrated.

Thank you very much for the question, Microglial cGAS-deficient mice have reduced infarct size, in this study, we supposed that A151 exert its protective role via cGAS pathways and microglial pyroptosis, thus no further reduced infarct size was observed in A151 treated cGAS-deficient mice.

We will perform double immunofluorescent staining of cGAS/Iba-1, cGAS/GFAP, and cGAS/NeuN to demonstrate the elimination of cGAS immunostaining from microglia and its overall reduction in microglial cGAS KO mice.

Further points:

- ds DNA staining shown in Figure 1 does not colocalize with DAPI and is not seen in the contralateral side. What type of changes in DNA structure would this antibody recognize given that the staining is specific? Also, microglia only occasionally show dsDNA staining, while several non-microglial cells do.

Thank you very much for the suggestion, As seen in figure 1, although dsDNA staining of the contralateral hemisphere was limited, larger amounts of dsDNA were found to accumulate in the ischemic hemisphere, and the immunostaining scale of dsDNA is larger than DAPI, which may indicate a leakage into cytoplasm of dsDNA.

Also, dsDNA staining were mainly shown in microglia, and only few in non-microglial cells. We will reconfirm this.

- Figure 1E: β -actin levels are highly variable making the interpretation of STING and cGAS western blotting results difficult.

Thank you very much for the suggestion, we will revise Figure 1E.

- Figure 3A: Immunostaining against IL-1 β and caspase-1 appears non-specific with high background. Please provide appropriate controls.

Thank you very much for the suggestion, we will revise Figure 3A and also provide appropriate controls.

- Figure 6: What type of cells show Annexin V and PI signal 3 days after MCAO? If these are neurons, and there is substantial delayed injury-related neuronal death 3 days after MCAO in these studies is there a significant increase in infarct size between 24h and 3 days post-stroke?

Thank you very much for the question. Actually, according to our Tunel/NeuN immunofluorescence staining, we observed less positive cell suggesting less neuronal death in A151-treated group than vehicle-treated group.

We will then measure infarct volumes through TTC staining of brain sections after MCAO, followed by reperfusion for 24 h, and we will compare whether there is a significant increase in infarct size between 24h and 3 days post-stroke.

- Did the authors check whether 4 weeks after tamoxifen treatment was sufficient for peripheral Cx3cr1-positive monocytes/macrophages to be replaced by other cells? Splenic cGAS expression in Cx3Cr1-positive monocytes should be shown.

Thank you for the suggestion, actually, we have assessed the relative expression of cGAS in microglia and splenic monocytes from cGAS-KO mice and wild-type mice, assessed using qPCR and western blot. We assessed microglial cGAS expression in cGAS-KO mice and found that both cGAS mRNA and protein levels were downregulated. While cGAS expression in splenic monocytes, however, was not downregulated.

- Is there any data showing brain penetration or pharmacokinetics of A151 after intraperitoneal injection?

Thank you very much for the question. In vivo, ODN A151 have been demonstrated to be effective to inhibit the development of organ-specific autoimmune diseases, such as arthritis and experimental autoimmune encephalomyelitis, which may indicate that A151 is a potent bioavailable, blood-brain barrier permeable, and nontoxic small molecule inhibitor [15,16].

To further clarify this issue, we will use immunofluorescent-tag labeled A151 to trace the brain penetration of A151 after intraperitoneal injection.

Thank you very much for the question.

Referee #3 (Comments on Novelty/Model System for Author):

1. In vivo model: long term outcomes are completely missing.

Thank you very much for the suggestion, we will assess the long-term outcome until 21 days, we determine to assess the neurological deficit and sensorimotor at the time points of 1, 3, 7, 14, 21 days.

2. In vitro model: the use of BV2 cell line is inadequate.

Thank you very much for the suggestion, we will perform experiments using primary cultures to confirm the results from BV2 cells, and detailed procedures were listed below.

Referee #3 (Remarks for Author):

This is a very interesting study that tested an interesting hypothesis regarding the role of dsDNA-induced cGAS and AIM2 pro-inflammatory pathways in a murine model of focal stroke and reperfusion. The authors also showed that administration of A151, a synthetic inhibitor for the aforementioned pathways conferred neuroprotection against ischemic injury.

While the study appears to be well designed and the ideas are interesting, there are many flaws in the data, which reduce the enthusiasm.

Main points:

1. A151 is synthetic oligonucleotides. In order to reach the targets, the systemically administrated A151 (i.p.) would have to go across the blood brain barrier, get into the brain parenchyma, and accumulate in the microglia with certain concentrations. The authors need to present evidence that A151 is indeed capable of doing all these. By the way, how did the authors determine the dose they used in the study?

Thank you very much for the question. In order to prove that A151 could go across the blood brain barrier, get into the brain parenchyma, and accumulate in the microglia with certain concentrations, we determine to use immunofluorescent labeled A151 and the fluorescence will be detected by certain machine.

We determined the dose of A151 according to previous studies, a dose that has been demonstrated to be effective in inhibiting immune activation in various organ-specific autoimmune diseases, such as experimental autoimmune encephalomyelitis (EAE)[15,16], collagen-induced arthritis[15], lipopolysaccharide (LPS)-induced toxic shock [17], and lupus nephritis [18].

2. As the authors correctly stated, the effect of microglial activation after brain ischemia is perhaps mainly the secondary brain injury. Unfortunately the stroke outcomes were assessed up to 3 days after ischemia. Long term outcomes are completely missing.

Thank you very much for the suggestion, it is very important. We will assess the long-term outcome until 21 days, we determine to assess the neurological deficit and sensorimotor at the time points of 1, 3, 7, 14, 21 days.

3. The quality of some key data is questionable. For instance, most Western blots were bleached, and many bands were oversaturated. The problems can be seen in Figure 1E, Figure 2B, Figure 2E, Figure 4A and Figure 4C.

Thank you very much for these suggestions, we will revise and provide high quality Western blots images.

4. All in vitro experiments were done using the BV2 cell line. This cell line has many problems. To the least, results from BV2 cells would need to be confirmed using primary cultures.

Thank you very much for the suggestion, we will perform experiments using primary cultures to confirm the results from BV2 cells.

We have successfully prepared mouse primary microglia by using brains of 1 day old mice, as previously described [19]. The resuspended cells were seeded in 75 cm² culture flasks and were maintained at 37°C in a cell culture chamber for 8–10 d before the microglia were separated from the mixed glial cell cultures by shaking the flasks at 260 rpm for 8 h at 37°C. The resuspended microglia were then seeded into 6-well plates at a density of 5×10⁵/cm² and cultured for 2 to 3 d before harvesting for experimental treatments. The purity of the microglia were over 90%, as determined by flow cytometry analysis.

Then we will perform experiments using primary cultures to confirm the results from BV2 cells. The microglia will be exposed to A151 (3 μM) for 4h in cell culture before being treated with LPS derived from *Escherichia coli* O55:B5 (Sigma-Aldrich) at a dose of 500 ng/ml LPS for 6 h, followed by stimulation with cytosolic double-stranded DNA (1 μg/ml). Prior to experimentation, the DNA was complexed with a transfection reagent to facilitate intracellular delivery. Lysates and supernatant from microglia will be subjected to western blot analysis, and LDH analysis. Immunostainings will also be performed to confirm the results of that in vitro [20].

Relatively minor points:

1. Figure 1 C, not sure that dsDNA is cytosolic in Iba-1 cells as claimed, as the dsDNA staining appears to co-localized with DAPI. Also, some GFAP+ cells also contain dsDNA.

Thank you very much for pointing this problem out, dsDNA mainly exists in microglia, only few in GFAP positive astrocytes. Cytoplasm dsDNA staining will be confirmed using a high resolution confocal laser microscope.

2. Figure 1D. Again, the majority of dsDNA staining appears to be nuclear, with close spatial co-localization with DAPI and 53BP1.

Thank you very much for pointing this problem out, the scale of dsDNA immunostaining is larger than DAPI and 53BP1. In terms of this issue, we will further confirm the cytoplasm dsDNA staining by using a high resolution confocal laser microscope.

3. Figure 2. Unclear where the tissues were sampled from the stroke brains for the various assays. Not sure why A151 reduced the protein complexes of AIM2 and ASC, and caspase-1 at both mRNA and protein levels.

Mice were sacrificed at day 3 after reperfusion, and cerebral tissues from ipsilateral hemispheres were harvested to extract RNAs and proteins.

Pyroptosis is closely related to inflammasome activation, is a programmed cell death mediated by caspase-1 [11]. In addition to the post-translational assembly of inflammasome that is directly triggered by dsDNA, type I interferons stimulated by the cGAS-STING pathway has been shown to be capable of enhancing the AIM2 inflammasome response by providing inflammasome-priming signals [21,22,23], and promoting the expression of inflammasome platforms and substrates [21,24], in the present study, we also observed that the synthetic oligodeoxynucleotides, cGAS inhibitor, A151, comprised of the immunosuppressive motif TTAGGG, reduced the protein complexes of AIM2 and ASC, and caspase-1 at both mRNA and protein levels.

4. Figure 3A. Not sure if any evidence for pyroptosis. The authors claimed that the GSDMD was found at cell plasma membrane; therefore this was consistent with

pyroptosis. However, this is hardly a supporting evidence for such a conclusion. The images could not tell whether it's cytosolic or cell membrane staining. No evidence was provided to indicate whether these GSDMD positive Iba-1 cells were live or dead/dying.

Thank you very much for pointing this problem. GSDMD has been demonstrated as a pore-forming membrane protein during pyroptosis, forming a “ring of fire”. In addition, morphological evidence of caspase-1 and GSDMD immunoreactivity were consistent with that observed in the CNS in both MS and EAE[12], thus we claimed that the GSDMD was found at cell plasma membrane; therefore this was consistent with pyroptosis.

Given that GSDMD acts as a pore-forming membrane protein during pyroptosis, and pyroptosis is an inflammatory programmed cell death, GSDMD positive Iba-1 cells may indicate a form and a progress of cell death, these GSDMD positive Iba-1 cells may be damaged and dying cells. [12].

We also performed double immunofluorescence staining (Figure 1), and the Iba-1 expression diminished with increasing GSDMD expression.

Figure 1

References

- [1] M. Gelderblom, A. Weymar, C. Bernreuther, J. Velden, P. Arunachalam, K. Steinbach, E. Orthey, T.V. Arumugam, F. Leypoldt, O. Simova, Neutralization of the IL-17 axis diminishes neutrophil invasion and protects from ischemic stroke, *Blood* 120 (2012) 3793-3802.
- [2] N.D. Kim, A.D. Luster, The role of tissue resident cells in neutrophil recruitment, *Trends in immunology* 36 (2015) 547-555.
- [3] J.-K. Strecker, A. Schmidt, W.-R. Schäbitz, J. Minnerup, Neutrophil granulocytes in cerebral ischemia—evolution from killers to key players, *Neurochemistry international* 107 (2017) 117-126.
- [4] P.M. Wise, D.B. Dubal, M.E. Wilson, S.W. Rau, M. Böttner, K.L. Rosewell, Estradiol is a protective factor in the adult and aging brain: understanding of mechanisms derived from in vivo and in vitro studies, *Brain Research Reviews* 37 (2001) 313-319.
- [5] S.-H. Yang, J. Shi, A.L. Day, J.W. Simpkins, Estradiol exerts neuroprotective effects when administered after ischemic insult, *Stroke* 31 (2000) 745-748.

- [6] D. Li, W. Lang, C. Zhou, C. Wu, F. Zhang, Q. Liu, S. Yang, J. Hao, Upregulation of microglial zeb1 ameliorates brain damage after acute ischemic stroke, *Cell reports* 22 (2018) 3574-3586.
- [7] M. Li, Z. Li, Y. Yao, W.-N. Jin, K. Wood, Q. Liu, F.-D. Shi, J. Hao, Astrocyte-derived interleukin-15 exacerbates ischemic brain injury via propagation of cellular immunity, *Proceedings of the National Academy of Sciences* 114 (2017) E396-E405.
- [8] H. Ren, Y. Kong, Z. Liu, D. Zang, X. Yang, K. Wood, M. Li, Q. Liu, Selective NLRP3 (pyrin domain-containing protein 3) inflammasome inhibitor reduces brain injury after intracerebral hemorrhage, *Stroke* 49 (2018) 184-192.
- [9] C. Qin, W.-H. Fan, Q. Liu, K. Shang, M. Murugan, L.-J. Wu, W. Wang, D.-S. Tian, Fingolimod protects against ischemic white matter damage by modulating microglia toward M2 polarization via STAT3 pathway, *Stroke* 48 (2017) 3336-3346.
- [10] J. Barrington, E. Lemarchand, S.M. Allan, A brain in flame; do inflammasomes and pyroptosis influence stroke pathology?, *Brain Pathology* 27 (2017) 205-212.
- [11] P. Broz, V.M. Dixit, Inflammasomes: mechanism of assembly, regulation and signalling, *Nature Reviews Immunology* 16 (2016) 407.
- [12] B.A. McKenzie, M.K. Mamik, L.B. Saito, R. Boghazian, M.C. Monaco, E.O. Major, J.-Q. Lu, W.G. Branton, C. Power, Caspase-1 inhibition prevents glial inflammasome activation and pyroptosis in models of multiple sclerosis, *Proceedings of the National Academy of Sciences* 115 (2018) E6065-E6074.
- [13] Y. Fu, Q. Liu, J. Anrather, F.-D. Shi, Immune interventions in stroke, *Nature Reviews Neurology* 11 (2015) 524.
- [14] A. Ahmad, M.M. Khan, M.N. Hoda, S.S. Raza, M.B. Khan, H. Javed, T. Ishrat, M. Ashafaq, M.E. Ahmad, M.M. Safhi, Quercetin protects against oxidative stress associated damages in a rat model of transient focal cerebral ischemia and reperfusion, *Neurochemical research* 36 (2011) 1360-1371.
- [15] L. Dong, S.i. Ito, K.J. Ishii, D.M. Klinman, Suppressive oligonucleotides protect against collagen-induced arthritis in mice, *Arthritis & Rheumatism* 50 (2004) 1686-1689.
- [16] P.P. Ho, P. Fontoura, P.J. Ruiz, L. Steinman, H. Garren, An immunomodulatory GpG oligonucleotide for the treatment of autoimmunity via the innate and adaptive immune systems, *The Journal of Immunology* 171 (2003) 4920-4926.
- [17] H. Shirota, I. Gursel, M. Gursel, D.M. Klinman, Suppressive oligodeoxynucleotides protect mice from lethal endotoxic shock, *The Journal of Immunology* 174 (2005) 4579-4583.
- [18] L. Dong, S. Ito, K.J. Ishii, D.M. Klinman, Suppressive oligodeoxynucleotides delay the onset of glomerulonephritis and prolong survival in lupus-prone NZB× NZW mice, *Arthritis & Rheumatism: Official Journal of the American College of Rheumatology* 52 (2005) 651-658.
- [19] J. Saura, J.M. Tusell, J. Serratos, High-yield isolation of murine microglia by mild trypsinization, *Glia* 44 (2003) 183-189.
- [20] F. Steinhagen, T. Zillinger, K. Peukert, M. Fox, M. Thudium, W. Barchet, C. Putensen, D. Klinman, E. Latz, C. Bode, Suppressive oligodeoxynucleotides

- containing TTAGGG motifs inhibit cGAS activation in human monocytes, *European journal of immunology* 48 (2018) 605-611.
- [21] F. Martinon, A. Mayor, J. Tschopp, The inflammasomes: guardians of the body, *Annual review of immunology* 27 (2009) 229-265.
- [22] L.I. Labzin, M.A. Lauterbach, E. Latz, Interferons and inflammasomes: cooperation and counterregulation in disease, *Journal of Allergy and Clinical Immunology* 138 (2016) 37-46.
- [23] F. Liu, Q. Niu, X. Fan, C. Liu, J. Zhang, Z. Wei, W. Hou, T.-D. Kanneganti, M.L. Robb, J.H. Kim, Priming and activation of inflammasome by canarypox virus vector ALVAC via the cGAS/IFI16–STING–type I IFN pathway and AIM2 sensor, *The Journal of Immunology* 199 (2017) 3293-3305.
- [24] J. Lugin, F. Martinon, The AIM 2 inflammasome: Sensor of pathogens and cellular perturbations, *Immunological reviews* 281 (2018) 99-114.

2nd Editorial Decision

9th October 2019

Thank you for your message asking us to reconsider our decision regarding your manuscript EMM-2019-11002. I have read your preliminary point-by-point response and have also sought external advice on the study from an expert in the field. Based on the outline you provide and the advice from the expert, we think that the proposed revisions sound reasonable. As such, we would invite you to submit a revised version of your study.

The advisor also provided comments and suggestions (see below) regarding your revision proposal, which we would strongly encourage you to also address:

1. We will perform apoptosis flow-cytometric analyses of microglia and proliferation of microglia as revealed by FACS analysis of Ki-67-stained cells in vehicle-treated and A151 treated mice after tMCAO to clarify these issues. LDH analysis or placental blue staining will be performed to clarify the microglial loss in vitro studies.

The advisor's comment:

The proposed measurements would make interpretation of the data easier. However, FACS will not reveal the relationship between microglial cell death / migration / proliferation and brain injury in vivo. The authors should also perform detailed and unbiased histological assessment of microglial cell death and proliferation using brain sections from mice 24h and 3 days after MCAO. Low resolution images showing the effect of A151 on microglial responses should be presented from the core and peri-infarct tissues with inserts showing colocalization of microglia with markers of proliferation and cell death in high resolution.

2. Thank you very much for the question. Inflammation and immune responses have emerged as important elements in the onset and progression of stroke, microglia play important role in inflammation[13]. By 3 days, microglia reach their peak number and peak inflammatory response in the peri-ischemic region, thus we chose this peak point to assess the effect of A151 on microglial pyroptosis. But this

does not imply that microglial pyroptosis as assessed at 3 days reperfusion is not a crucial contributor to disease pathophysiology.

We will perform TTC staining to acquire the 24 h infarct size data after both A151 treatment and in microglial cGAS deficient mice.

The advisor's comment:

The authors should also statistically compare infarct size between mice with 24h and 3days survival.

3. Thank you very much for the suggestion, we will redo the analysis to acquire the exact infarct size. The total infarct volume for the entire brain will be calculated as the sum of the infarct volume of each section. To compensate for the effect of brain edema, the infarct volume percentage was calculated as follows: $\text{infarct area} \times (1 - [(\text{ipsilateral hemisphere area} - \text{contralateral hemisphere area}) / \text{contralateral hemisphere}])$. Infarct volumes were calculated by the integration of infarcted areas on each brain slice, as quantified with a computer-assisted image analyzer [14]. Individual data points will also be provide for given mice on all graphs reporting infarct volume.

The new formula for the infarct volume correct the edema, and we will also compare the brain edema to show whether it differs between vehicle and A151-treated mice.

The advisor's comment:

I still do not understand how the measured infarct size could have derived from the images presented due to a large difference in the visible lesion. Please provide data from individual mice.

4. Thank you very much for the question, Microglial cGAS-deficient mice have reduced infarct size, in this study, we supposed that A151 exert its protective role via cGAS pathways and microglial pyroptosis, thus no further reduced infarct size was observed in A151 treated cGAS-deficient mice.

We will perform double immunofluorescent staining of cGAS/Iba-1, cGAS/GFAP, and cGAS/NeuN to demonstrate the elimination of cGAS immunostaining from microglia and its overall reduction in microglial cGAS KO mice.

The advisor's comment:

Not sufficient to resolve this issue. The weakness of this experiment should also be at least discussed as pointed out above. It is not expected that infarct volume around 10-15% of the hemisphere can be reduced further in the experimental model.

5. Thank you very much for the suggestion, As seen in figure 1, although dsDNA staining of the contralateral hemisphere was limited, larger amounts of dsDNA were found to accumulate in the ischemic hemisphere, and the immonostaining scale of dsDNA is larger than DAPI, which may indicate a leakage into cytoplasm of dsDNA.

Also, dsDNA staining were mainly shown in microglia, and only few in non-microglial cells. We will reconfirm this.

The advisor's comment:

Please also provide high resolution images of individual microglial cells where any leakage of dsDNA into the cytoplasm can be appreciated.

All the other issues raised by the reviewers would need to be convincingly addressed. Please note that the revised manuscript needs to be reviewed.

1st Revision - authors' response

7th January 2020

***** Reviewer's comments *****

Referee #1 (Comments on Novelty/Model System for Author):

Only young, male mice are used and only short-term outcomes are assessed, which limit the medical impact.

Thank you very much for the suggestion, we have also used aged mice (12 months) to assess the therapeutic efficacy of A151. We have assessed the long-term outcome until 14 days to evaluate the neurological deficits and sensorimotor using aged mice at the time points of 1, 3, 7, and 14 days.

As shown in Figure S1. A151 attenuates brain injury in aged mice at 24 h after MCAO (Page 18). Neurological deficits (A) including mNSS score, corner turning test, foot-fault test, and adhesive-removal test. n = 10 per group. (B) Representative TTC staining image and (C) quantification of infarct lesions 24 h after MCAO in vehicle- and A151 treated aged mice until 14 days, indicating that A151 may be superior in elderly patients who do not just have brain ischemic disease, but also have a compromised immune system.

Figure S1

As for young mice, as shown in Figure 6. Neurological tests were carried out to evaluate the motor, sensory, and balance functions in mice receiving A151, or vehicle at days 1, 3, 7, and 14 after MCAO. Improved neurodeficits persist in A151 treated mice until day 14 after brain ischemia. A151 treatment reduced the severity of neurobehavioral deficits assessed by mNSS, corner-turning test, foot-fault test, and adhesive removal test as early as at day 1 after MCAO and persisted to day 14 after MCAO (Fig. 6B) (Page 18), suggesting that inhibition of cytoplasm dsDNA-sensing cGAS A151 can provide long-term benefit after MCAO.

Referee #1 (Remarks for Author):

The authors provide the results of a series of experiments aimed to determine the therapeutic efficacy of inhibiting cytosolic dsDNA on inflammation and outcomes after experimental cerebral ischemia. They utilize immunofluorescence, Western blotting, PCR, in vivo and in vitro models, and transgenic mice to confirm the role of cGAS in post-stroke inflammation. Therapies targeting post-stroke inflammation represent an important potential treatment for stroke, and thus the work is considered of potentially high impact.

The activation of cGAS is attributed to microglia but blood-derived macrophages have infiltrated the area by day 3 and Iba1 cannot differentiate the cell types. Were there differences in cytokine production by flow cytometry within the macrophages as well? Only the microglia data is shown in Figure 5.

Thank you very much for the question. We have done double immunofluorescence staining of cGAS/F4/80 (blood-derived macrophages marker) to differentiate the cell types of microglia and blood-derived macrophages.

Figure S2

As shown in Figure S2A. Macrophages also partly co-express cGAS in the penumbra of MCAO animals. Immunofluorescent labeling of penumbra from MCAO animals sacrificed at day 3. (A) Macrophage marker F4/80 (green) and cGAS (red) co-localize within the same cells (White arrows).

As shown in Figure S2B and C. Flow Cytometry analysis has been performed using the following fluorochrome-conjugated mouse reactive antibodies against CD45, CD11b, F4/80, interleukin-6 (IL-6), IL-4, TNF- α , and TGF- β to measure cytokine production within the macrophages.

Macrophages also co-express cGAS in the penumbra of MCAO animals (Fig. S2A), A151 also affects the production of IL-4 and TNF- α from blood-derived macrophages (Fig. S2B

and C), suggesting that A151 effectively inhibits the activation of microglia/macrophage in the brain after MCAO. (Page 17)

They have an interesting finding of reduced neutrophil infiltration in the A151-treated mice. What cells produce neutrophil chemokines in the brain after ischemia and were these suppressed by A151 or could there be a direct effect on A151 on neutrophils?

Thank you very much for the question, the recruitment of neutrophils into tissue is orchestrated by tissue-resident cells such as astrocytes and microglia, which could release and the main source of multiple chemokines or adhesion molecules [1,2,3], such as CCL2, CCL5, CXCL1, CXCL5, and CXCL10, thus we performed PCR to analyze the expression level of these genes. Double immunofluorescence staining of ly6G/cGAS has also been performed to clarify whether there is a direct effect on A151 on neutrophils.

Figure S3

A151 reduces neutrophil chemoattractants CCL2 and CXCL10 (Fig. S3A), this selective migration of neutrophils might due to the suppressed neutrophil-attracting chemokines that that mainly recruits neutrophils by A151. A151 may also have a direct effect on neutrophils since several cells expressing cGAS were also positive for ly6G of neutrophils marker (Fig S3B) (Page 17).

Several cells expressing cGAS were also positive for ly6G of neutrophils or F4/80 of macrophage marker, these results suggest that the protection of A151 may not entirely depend on brain resident microglia. Nevertheless, given the widespread distribution of cGAS on multiple types of cells including neutrophils, regulation of cGAS on microglia is the main but not the sole mechanism responsible for the protection offered by A151, the operating mechanisms of A151 on other potential cellular targets in MCAO require future investigation, we mentioned this in the discussion (Page 21, 22).

Several requirements for preclinical stroke modeling are missing from the methods. Were sample sizes prespecified? Were mice randomized to treatment with A151? How? They should justify the use of only young, male mice.

Thank you very much for the question. Yes, sample sizes were prespecified. We prespecified the sample size per group by power analysis using a significance level of $\alpha=0.05$ at $\beta=0.2$ with 80% power to detect statistical differences. The results of sample size calculation show that $n = 6-10$ each group of mice is appropriate. SAS 9.1 software (SAS Institute Inc, Cary, NC) was used for power analysis and sample size calculations (Page 13). Moreover, the number of mice in each group is generally determined according to the type of experiment, such as neurological scoring, the variation among mice in the group is large, so the number is generally large, about 10 mice per group, while the number of mice used in quantitative index analysis such as flow cytometry is relatively small, about 4-6

mice per group, which were also based on the previous studies [4,5,6], and are approved and recognized in the field. The mice were randomized to treatment with A151.

Thank you very much for the question. Actually, previous studies have used various mice types to make models of middle cerebral artery occlusion including female mice, male mice, young mice, and aged mice. Estradiol is a protective factor in the adult and aging brain against cerebral ischemia [7,8]. Moreover, according to epidemiological data, incidence of ischemic stroke increases with age, which occurs frequently in middle-aged and elderly people, and is higher in men than in women. Mice at 8 weeks are middle-aged mice, thus we chose 8 weeks male mice to do the experiment, which is widely used in the research field of ischemic stroke [5,9].

As for the aged mice, it is highly necessary and important to use aged mice (12 months) to evaluate the role of inhibition of cGAS in supplementary experiments, since ischemic stroke is more common in the elderly. We have assessed the long-term outcome until 14 days to evaluate the neurological deficits and sensorimotor using aged mice at the time points of 1, 3, 7, and 14 days (Figure S1) (Page 18).

Only a single time point (day 3) is shown. Does the inhibition of cGAS lead to long term improvements in outcome?

Thank you very much for the suggestion, it is very important. We have also assessed the long-term outcome until 14 days to assess the neurological deficit and sensorimotor to determine the role of inhibition of cGAS via A151 at the time points of 1, 3, 7, and 14 days.

As shown in Figure S1. A151 attenuates brain injury in aged mice at 24 h after MCAO (Page 18). Neurological deficits (A) including mNSS score, corner turning test, foot-fault test, and adhesive-removal test. n = 10 per group. (B) Representative TTC staining image and (C) quantification of infarct lesions 24 h after MCAO in vehicle- and A151 treated aged mice until 14 days, indicating that A151 may be superior in elderly patients who do not just have brain ischemic disease, but also have a compromised immune system.

As for young mice, as shown in Figure 6. Neurological tests were carried out to evaluate the motor, sensory, and balance functions in mice receiving A151, or vehicle at days 1, 3, 7, and 14 after MCAO. Improved neurodeficits persist in A151 treated mice until day 14 after brain ischemia. A151 treatment reduced the severity of neurobehavioral deficits assessed by mNSS, corner-turning test, foot-fault test, and adhesive removal test as early as at day 1 after MCAO and persisted to day 14 after MCAO (Fig. 6B) (Page 18), suggesting that inhibition of cytoplasm dsDNA-sensing cGAS A151 can provide long-term benefit after MCAO.

Minor English grammar errors require editing.

Thank you for the suggestion, we have edited English grammar errors.

Referee #2 (Comments on Novelty/Model System for Author):

A potentially interesting paper, but there are major conceptual issues that limit my enthusiasm in its present form. Key issues are the missing evidence for the role of microglial pyroptosis in brain injury in this model and the assessment of exclusively delayed inflammatory responses that weaken the authors claims. Please refer to the corresponding details and other points in my assessment.

Thank you very much for the suggestions, as for the role of microglial pyroptosis in brain injury in this model, one of the defining features of inflammasome activation is the induction of a pro inflammatory programmed cell death termed pyro- (fire/fever) ptosis (falling) [10]. It has been demonstrated that the canonical inflammasome pathway aggravates brain ischemic injury through the spread of inflammation after stroke, while inhibiting the inflammasome pathway activation provides protective effects in experimental stroke models [10]. Pyroptosis is an active caspase-1 mediated pro-inflammatory lytic programmed cell death through cleavage of gasdermin D (GSDMD). The GSDMD forms membrane pores, which causes cell swelling and subsequently amplify the inflammation through the concomitant release of neurotoxic and inflammatory mediators [11], indicating that pyroptosis is a potential target for ischemic stroke intervention.

Moreover, according to the suggestions, we also assessed the accumulation of dsDNA and the inflammatory responses at early time points of 6 h and 24 h (Fig S4) (Page 15).

Referee #2 (Remarks for Author):

In the research paper by Li et al., the authors show that the Cyclic GMP-AMP synthase (cGAS) antagonist A151 reduces brain injury and inflammation after experimental stroke in mice. They find that administration of A151 dampens the expression of cGAS, AIM2, caspase-1, gasdermin D and IL-1 β /IL-18 in the ischemic brain tissue 3 days after stroke, which parallels reduced microglial numbers, microglial cytokine production and neutrophil counts. Microglial cGAS deficiency is also associated with smaller infarct size and improved neurological outcome. Based on these data, the authors conclude that cytosolic DNA sensing via cGAS and AIM2 collectively amplify inflammatory responses during brain ischemia and inhibition of dsDNA-sensing cGAS is sufficient to attenuate inflammation and brain injury, providing novel therapeutic opportunities for ischemic stroke.

Novel discoveries in the paper include the protective role of A151 in experimental stroke and that brain injury is reduced in the absence of microglial cGAS in mice. These results are potentially interesting for the field. There are, however several major conceptual and technical issues that limit the impact of the present studies.

Specific points:

- The authors emphasize the role of dsDNA sensing cGAS in dampening inflammation as being a crucial mechanism for brain protection. To justify this conclusion, dsDNA staining is performed and microglia, astrocytes, inflammasome-related molecules and inflammatory cytokines are measured as late as 3 days after stroke. Since based on a number of studies, blockade of inflammatory mediators (including inflammasome- and IL-1-related pathways) is primarily effective in the first hours up to one day after acute brain injury when proinflammatory cascades are induced, it is not possible to draw appropriate conclusions from this data set regarding the mechanisms through which cGAS inhibition reduces brain injury. At 3 days anything measured in the brain may be secondary to smaller infarct size, while it remains unclear if inflammatory mediators are altered between 4h and 24h after stroke when the majority of the infarct is formed. In line with this, levels of IL-1 β and several downstream inflammatory cytokines/chemokines (IL-6, MCP-1, etc) are known to peak between 4h and 24h after cerebral ischemia in mice. The authors must provide comparative assessment of inflammasome activation and microglial responses at earlier time points (ideally at 4-8h and 24h post-stroke) to support their conclusions. Similarly, dsDNA staining at 3 days post-stroke may simply indicate the presence of heavily infarcted tissue and it remains unclear whether similar increases are seen at earlier time points to influence microglial inflammatory responses at the critical time window.

Thank you very much for this useful suggestion. We have performed PCR analysis of AIM2, ASC, caspase-1, IL-1 β , IL-6, and MCP-1 to provide a comparative assessment of inflammasome activation and microglial responses in vehicle-treated and A151 treated group at earlier time points of 6 h and 24 h after stroke.

As shown in Figure S4. Administration of A151 dampens the mRNA expression of AIM2 inflammasome related molecules (AIM2/Caspase-1/ASC), levels of IL-1 β , and several downstream inflammatory cytokines/chemokines (IL-6, MCP-1) (Fig S4), suggesting the anti-inflammatory effect of A151 treatment initiated after the onset of cerebral ischemia (Page 15).

Figure S4

Immunofluorescence staining of dsDNA have been performed to clarify whether similar increases can be seen at earlier time points of 6 h and 24 h after stroke to influence microglial inflammatory responses.

As shown in Figure 1A, similar increases and the leakage of dsDNA into the cytoplasm were also detected at earlier time points of 6 h and 24 h after MCAO (Page 13 and 14).

- The authors suggest that dsDNA triggers microglial pyroptosis, which is prevented by A151 leading to reduced brain injury. They show that microglia are GSDMD-positive and conclude that microglia are the main cells undergoing pyroptosis after MCAO. There several problems with these interpretations. First, the authors do not show microglial cell death with specific markers. Second, they show by using both flow cytometry and histology that A151 treatment is associated with less microglia in the brain 3 days after stroke (Figure 5). If microglia die and A151 reverses this, one would expect more microglia to survive. If the authors claim (although they do not prove) that microglial pyroptosis is important in disease pathophysiology in the current model, they should at least follow microglial numbers over time in both the infarct core and the peri-infarct tissues and assess cells death and proliferation simultaneously to make these claims sound. In Figure 5H, morphologically activated microglial cells are shown and these are reduced in numbers in A151-treated mice, which raises the possibility that A151 actually increases microglial cell death or reduces their proliferation/migration. Did the authors observe any microglial loss in their in vitro studies?

Thank you very much for the question, in the present study, we examined whether pharmacologically antagonizing dsDNA-sensing cGAS via A151 could govern pyroptosis and the overall neuroinflammation in the context of brain ischemia. Herein, we provide evidence for the activation and formation of pyroptosis following stroke, the effector molecular GSDMD along with the caspase-1 immunoreactivity accumulated on the plasma membrane, forming a distinctive “ring of fire” morphology in the ischemic brain (Fig 3), consistent with pyroptosis observed in a previous study in EAE brain tissue sections[12]. Moreover, A151 effectively reduced the expression of cGAS, AIM2 inflammasome, and pyroptosis related molecules, including caspase-1, gasdermin D, IL-1 β , and IL-18.

As for the number of microglia, the number of microglia = microglial proliferation/migration - microglial cell death. However, this may not mean that A151 actually increases microglial cell death or reduces their proliferation/migration, the final microglia number depends on the degree of both microglial cell death and their proliferation/migration. The number of microglia $\downarrow\downarrow$ = their proliferation/migration $\downarrow\downarrow$ - microglial cell death \downarrow , which may also lead to a less number of the microglia.

We have performed double immunofluorescence staining for microglial proliferation and apoptosis as revealed by Iba-1/ Ki-67 and Iba-1/Tunel in the peri ischemic area as well as in the ischemic core in vehicle-treated and A151 treated mice after tMCAO to clarify these issues.

Double immunofluorescence staining for microglial proliferation and apoptosis as revealed by Iba-1/ Ki-67 and Iba-1/Tunel were performed in the peri-ischemic area as well as in the ischemic core in vehicle- and A151 treated mice after MCAO, A151 reduces microglial cell death and proliferation after stroke (Fig. S5) (Page 17).

As shown in Figure S5. A151 reduces microglial cell death and proliferation after stroke. (A and C) Representative double immunofluorescence stainings for apoptosis and proliferation of microglia in the penumbra and ischemic core in vehicle - and A151- treated MCAO mice using ki67 (left panel) and TUNEL staining (right panel). (B and D) Quantification of TUNEL/Iba-1 and ki67/Iba-1 double-positive cells.

Figure S5

Thank you very much for the question, MTS and LDH analysis were performed to clarify the microglial loss in vitro studies.

MTS assay and LDH analysis were performed to clarify the microglial loss in vitro studies. Primary microglia were treated with A151 or PBS. MTS values were normalized to PBS-treated controls and showed no difference between the two groups (Fig. S6). To quantify the microglial loss, lactate dehydrogenase (LDH) activity was measured in cell culture supernatants, indicating extravasation of cellular contents. Poly(dA:dT) exposure increased LDH release from microglia,

while A151 treatment reduced its release in Poly(dA:dT)-exposed cells (Fig. S6) (Page 16).

Figure S6

- Neurological assessment suggests that brain protection by A151 may be apparent already at 24 h post-stroke. Would this imply that microglial pyroptosis as assessed at 3 days reperfusion is not a crucial contributor to disease pathophysiology? To overcome this controversy, 24h infarct size data are strongly required after both A151 treatment and in microglial cGAS deficient mice.

Thank you very much for the question. Inflammation and immune responses have emerged as important elements in the onset and progression of stroke, microglia play important role in inflammation[13]. By 3 days, microglia reach their peak number and peak inflammatory response in the peri-ischemic region, thus we chose this peak point to assess the effect of A151 on microglial pyroptosis.

We have performed TTC stainings to acquire the 24 h infarct size data after both A151 treatment and in microglial cGAS deficient mice. Reduced 24 h infarct size data were observed after both A151 treatment and in microglial cGAS deficient mice, suggesting cGAS inactivation may reduce volumes of cerebral infarct in early stages of ischemic injury development.

As shown in Figure S8. A151 treatment and microglial cGAS inactivation attenuates ischemic brain injury at 24 h after MCAO (Page 18). Infarct volume determined by TTC staining. The red regions show intact areas; pale regions show infarct areas. (A) TTC-stained brain slices showing the infarct areas (dark dashed lines) of mice receiving A151 or vehicle at 24 h after reperfusion. (B) Graph show the infarct volumes in vehicle- and A151-treated MCAO mice (n=7 mice per group). Infarct volumes were measured from TTC-stained brain sections. (C) Representative coronal sections from WT and cGAS-KO mice after 24 h of reperfusion following 60-min MCAO were stained with TTC. (D) Quantification of brain infarct volume (n=6).

Figure S8

-The TTC images presented in Figures 6C and 7F are not in line with the percentage infarct volume data as shown on the graphs. In fact, 60min MCAO should result in much larger injury than 20-30% of the ipsilateral hemisphere. Please give details how exactly infarct size was measured and provide individual data points for given mice on all graphs reporting infarct volume. Has oedema correction been used? Did brain oedema differ between vehicle and A151-treated mice?

Thank you very much for the suggestion, I am sorry that this part made you confused in the original manuscript. We have redone the analysis to acquire the exact infarct size. The total infarct volume for the entire brain will be calculated as the sum of the infarct volume of each section. The infarct volume for each brain were calculated by the integration of infarcted areas and the distance between slices, as quantified with a computer-assisted image analyze. We also calculated the oedema volume as follows: volume of the contralateral hemisphere - the volume of the ipsilateral hemisphere. To compensate for the effect of brain edema, the corrected infarct volume was calculated by using the following equation: $\text{infarct volume} \times (1 - [(\text{ipsilateral hemisphere volume} - \text{contralateral hemisphere volume}) / \text{contralateral hemisphere}])$ [14]. Individual data points have also been provide for given mice on graphs reporting infarct volume (Figures 6D). The new formula for the infarct volume correct the edema, and we have compared the brain edema to show whether it differs between vehicle and A151-treated mice. As shown in Figures 6D, A151 reduces infarct volume and brain edema in mice subjected to MCAO.

Individual data points have also been provide for given mice on graphs reporting infarct

volume for Figure 7G

- Microglial cGAS-deficient mice have reduced infarct size and based on the data presented it is unlikely that this could be reduced any further in this experimental model. Therefore, the lack of an effect of cGAS inhibition in these mice is difficult to interpret. The authors should revise related statements in the manuscript. Elimination of cGAS immunostaining from microglia and its overall reduction in microglial cGAS KO mice should be demonstrated.

Thank you very much for the question, to further determine the mechanism by which A151 exerts neuroprotective effects, cGAS-KO mice were used in the current study. We redo the analysis using the new formula for the infarct volume (Fig 7G). Microglial cGAS-deficient mice exhibited reduced infarct size, of interest, when cGAS-KO mice were treated with

vehicle or A151 after MCAO, these two groups did not differ in infarct volume, mNSS score, no further reduced infarct size was observed in A151 treated cGAS-deficient mice, suggesting that the beneficial effects of A151 treatment are diminished in transgenic mice with depletion of microglial cGAS, these findings suggest that the cGAS pathway is essential for A151 to exert its neuroprotective role in the context of brain ischemic injury. However, the weakness of this experiment may be that it is not expected that reduced infarct volume of microglial cGAS-deficient mice presented to be reduced further in the experimental model. The effect of cGAS inhibition in these mice is therefore difficult to interpret. We have mentioned this in the discussion section (Page 21).

Thank you for the suggestion, we have performed double immunofluorescent staining of cGAS/Iba-1 to demonstrate the elimination of cGAS immunostaining from microglia and western blot to show its overall reduction in microglial cGAS KO mice.

As shown in (Fig S7), we have assessed the relative expression of cGAS in microglia and splenic monocytes from cGAS-KO mice and wild-type mice, assessed using western blot. We assessed microglial cGAS expression in cGAS-KO mice and found that cGAS protein levels were downregulated. While cGAS expression in splenic monocytes, however, was not downregulated. Double immunofluorescent staining of cGAS/Iba-1 further demonstrated the elimination of cGAS immunostaining from microglia (Page 19).

Figure S7

Figure S7. (A) cGAS expression was assessed in microglia and monocytes by western blot. (B) Representative double immunofluorescence stainings for cGAS and Iba-1 in WT and cGAS-KO mice.

Further points:

- ds DNA staining shown in Figure 1 does not colocalize with DAPI and is not seen in the contralateral side. What type of changes in DNA structure would this antibody recognize given that the staining is specific? Also, microglia only occasionally show dsDNA staining, while several non-microglial cells do.

Thank you very much for the suggestion. We have revised Figure 1A and B with high magnification images, although dsDNA staining of the sham brain was limited and

distributed predominantly as ring-like nuclear contours, after the induction of MCAO, the intensities of dsDNA increased with a nucleoplasmic relocation in the penumbra, the leakage of dsDNA into the cytoplasm was also detected at time points of 6 h, 24 h, as well as 3 d after MCAO (Fig. 1A). Moreover, following stroke, the majority of dsDNA staining appears to be nuclear, with close spatial co-localization with DAPI and 53BP1, while cytoplasmic dsDNA could also be detected in the ischaemic penumbra (Fig. 1B). Since astrocytes and microglia are the main immune cells that quickly respond following ischemia and participate in the neuroinflammation, next, we performed double immunofluorescent analysis of dsDNA with cell type-specific markers for microglia (Iba1) and astrocytes (GFAP). Cytosolic dsDNA immunofluorescent signals could be detected in microglial Iba-1 and astrocytic GFAP (Fig. 1B) (Page 14).

- Figure 1E: β -actin levels are highly variable making the interpretation of STING and cGAS western blotting results difficult.

Thank you very much for the suggestion, we have revised Figure 1E.

- Figure 3A: Immunostaining against IL-1 β and caspase-1 appears non-specific with high background. Please provide appropriate controls.

Thank you very much for the suggestion, we have revised Figure 3A and provided appropriate controls (Figure 3A).

- Figure 6: What type of cells show Annexin V and PI signal 3 days after MCAo? If these are neurons, and there is substantial delayed injury-related neuronal death 3 days after MCAo in these studies is there a significant increase in infarct size between 24h and 3 days post-stroke?

Thank you very much for the question. Cells that show Annexin V and PI signal 3 days after MCAo could be neurons or microglia. According to our Tunel/Iba-1 immunofluorescence staining (Fig. S5C and D), we observed less Tunel/Iba-1 double positive cell suggesting less microglia death in A151-treated group than vehicle-treated group.

We have measured infarct volumes through TTC staining of brain sections after MCAO, followed by reperfusion for 24 h, and we compared whether there is a significant increase in infarct size between 24h and 3 days post-stroke. There is an increase in infarct size between 24h and 3 days, however, this did not reach a statistical significance.

- Did the authors check whether 4 weeks after tamoxifen treatment was sufficient for peripheral Cx3cr1-positive monocytes/macrophages to be replaced by other cells? Splenic cGAS expression in Cx3Cr1-positive monocytes should be shown.

Thank you for the suggestion, we have assessed the relative expression of cGAS in microglia and splenic monocytes from cGAS-KO mice and wild-type mice, assessed using

western blot. We assessed microglial cGAS expression in cGAS-KO mice and found that cGAS protein levels were downregulated. While cGAS expression in splenic monocytes, however, was not downregulated, thus indicating 4 weeks after tamoxifen treatment was sufficient for peripheral Cx3cr1-positive monocytes/macrophages to be replaced by other cells. Microglial cGAS expression in WT and cGAS-KO mice were also evaluated by immunofluorescence staining to further demonstrated the elimination of cGAS immunostaining from microglia.

Figure S7

Figure S7. (A) cGAS expression was assessed in microglia and monocytes by western blot. (B) Representative double immunofluorescence stainings for cGAS and Iba-1 in WT and cGAS-KO mice.

- Is there any data showing brain penetration or pharmacokinetics of A151 after intraperitoneal injection?

Thank you very much for the question. Firstly, this is a question that we have carefully considered before we chose the method of drug administration. According to previous studies, under physiological conditions, it has been shown that oligonucleotides in brain tissue could be detected, although the concentration is low [15,16]. Furthermore, A151 even in high concentration are non-toxic, a previous study has shown that administering A151 for up to 32 weeks is safe and does not adversely impact the health of mice [17]. Studies dealing with the intraperitoneal injection of ODNs of specific sequences in experimental mice stroke model and Alzheimer disease have been reported[18,19,20], indicating that the ODNs could get into the brain parenchyma, and accumulate and influence brain cells. In the present study, we observed that A151 reduced the expressions of both cGAS mRNA and protein levels in the ischemic brain following cerebral I/R. We presumed that systemic delivery of A151 can be absorbed from the blood into the damaged brain parenchyma via the disruption of the blood-brain barrier. Moreover, A151 may accumulate mainly in microglia, since microglia have phagocytic function. However, this study was now unable to provide direct evidence for the entry of A151 into brain tissue after cerebral infarction, because of its importance for determine the mechanism of immunomodulatory therapies, this information is critical and will be carefully explored in future studies.

Referee #3 (Comments on Novelty/Model System for Author):

1. In vivo model: long term outcomes are completely missing.

Thank you very much for the suggestion, we have also assessed long-term outcome until 14 days to evaluate the neurological deficits and sensorimotor using aged mice at the time points of 1, 3, 7, and 14 days.

As shown in Figure 6. Neurological tests were carried out to evaluate the motor, sensory, and balance functions in mice receiving A151, or vehicle at days 1, 3, 7, and 14 after MCAO. Improved neurodeficits persist in A151 treated mice until day 14 after brain ischemia. A151 treatment reduced the severity of neurobehavioral deficits assessed by mNSS, corner-turning test, foot-fault test, and adhesive removal test as early as at day 1 after MCAO and persisted to day 14 after MCAO (Fig. 6B) (Page 18), suggesting that inhibition of cytoplasm dsDNA-sensing cGAS A151 can provide long-term benefit after MCAO.

2. In vitro model: the use of BV2 cell line is inadequate.

Thank you very much for the suggestion, we have performed experiments using primary cultures to confirm the results from BV2 cells (Fig 4).

Referee #3 (Remarks for Author):

This is a very interesting study that tested an interesting hypothesis regarding the role of dsDNA-induced cGAS and AIM2 pro-inflammatory pathways in a murine model of focal stroke and reperfusion. The authors also showed that administration of A151, a synthetic inhibitor for the aforementioned pathways conferred neuroprotection against ischemic injury.

While the study appears to be well designed and the ideas are interesting, there are many flaws in the data, which reduce the enthusiasm.

Main points:

1. A151 is synthetic oligonucleotides. In order to reach the targets, the systemically administrated A151 (i.p.) would have to go across the blood brain barrier, get into the brain parenchyma, and accumulate in the microglia with certain concentrations. The authors need to present evidence that A151 is indeed capable of doing all these. By the way, how did the authors determine the dose they used in the study?

This is a question that we have carefully considered before we chose the method of drug administration. According to previous studies, under physiological conditions, it has been shown that oligonucleotides in brain tissue could be detected, although the concentration is low [15,16]. Furthermore, A151 even in high concentration are non-toxic, a previous study has shown that administering A151 for up to 32 weeks is safe and does not adversely impact the health of mice [17]. Studies dealing with the intraperitoneal injection of ODNs of specific sequences in experimental mice stroke model and Alzheimer disease have been reported[18,19,20], indicating that the ODNs could get into the brain parenchyma, and accumulate and influence brain cells. In the present study, we observed that A151 reduced the expressions of both cGAS mRNA and protein levels in the ischemic brain following cerebral I/R. We presumed that systemic delivery of A151 can be absorbed from the blood into the damaged brain parenchyma via the disruption of the blood-brain barrier. Moreover, A151 may accumulate mainly in microglia, since microglia have phagocytic function. However, this study was now unable to provide direct evidence for the entry of A151 into brain tissue after cerebral infarction, because of its importance for determine the mechanism of immunomodulatory therapies, this information is critical and will be carefully explored in future studies. In future studies, we should consider exploring carriers that A151 can enter brain tissue under normal circumstances, such as gold nanoparticles or liposome encapsulated polyethylenimine/ODN polyplexes for brain targeting, to enhance its effect.

We determined the dose of A151 according to previous studies, a dose that has been demonstrated to be effective in inhibiting immune activation in various organ-specific autoimmune diseases, such as experimental autoimmune encephalomyelitis (EAE)[21,22], collagen-induced arthritis[21], lipopolysaccharide (LPS)-induced toxic shock [23], lupus nephritis [17], and ocular inflammation [24], thus when A151 was delivered systemically, the effective dose in mice was typically 300 µg.

2. As the authors correctly stated, the effect of microglial activation after brain ischemia is perhaps mainly the secondary brain injury. Unfortunately, the stroke outcomes were assessed up to 3 days after ischemia. Long term outcomes are completely missing.

Thank you very much for the suggestion, we have assessed long-term outcome until 14 days to evaluate the neurological deficits and sensorimotor using aged mice at the time points of 1, 3, 7, and 14 days. As shown in Figure 6. Neurological tests were carried out to evaluate the motor, sensory, and balance functions in mice receiving A151, or vehicle at days 1, 3, 7, and 14 after MCAO. Improved neurodeficits persist in A151 treated mice until day 14 after brain ischemia. A151 treatment reduced the severity of neurobehavioral deficits assessed by mNSS, corner-turning test, foot-fault test, and adhesive removal test as early as at day 1 after MCAO and persisted to day 14 after MCAO (Fig. 6B) (Page 18), suggesting that inhibition of cytoplasm dsDNA-sensing cGAS A151 can provide long-term benefit after MCAO.

3. The quality of some key data is questionable. For instance, most Western blots were bleached, and many bands were oversaturated. The problems can be seen in Figure 1E, Figure 2B, Figure 2E, Figure 4A and Figure 4C.

Thank you very much for these suggestions, as shown in Figure 1E, Figure 2B, Figure 2E, Figure 4A and Figure 4C, we have revised and provided high quality western blots images.

4. All in vitro experiments were done using the BV2 cell line. This cell line has many problems. To the least, results from BV2 cells would need to be confirmed using primary cultures.

Thank you very much for the suggestion, we have performed experiments using primary cultures to confirm the results from BV2 cells (Fig 4). The expression levels of cGAS axis and pyroptosis associated proteins (caspase-1, IL-1 β , and GSDMD) were elevated in both lysates (Fig. 4A-B) and supernatants (Fig. 4C-D) of LPS-primed primary microglia stimulated with poly (dA:dT) compared with the blank control group, whereas the elevation of these proteins was suppressed by A151 (Page 16).

Relatively minor points:

1. Figure 1 C, not sure that dsDNA is cytosolic in Iba-1 cells as claimed, as the dsDNA staining appears to co-localized with DAPI. Also, some GFAP+ cells also contain dsDNA.

Thank you very much for pointing this problem out. Cytoplasm dsDNA staining have been confirmed in a high magnification image. As shown in Figure 1A, the expression of dsDNA increased following brain ischemia, and cytosolic dsDNA positive signals could be observed in microglia and also in GFAP positive astrocytes, we have modified this in Figure 1B (Page 13 and 14).

2. Figure 1D. Again, the majority of dsDNA staining appears to be nuclear, with close spatial co-localization with DAPI and 53BP1.

Thank you very much for pointing this problem out, we have revised this in Fig. 1B with high magnification images, following stroke, the majority of dsDNA staining appears to be nuclear, with close spatial co-localization with DAPI and 53BP1, while cytoplasmic dsDNA could also be detected in the ischaemic penumbra (Fig. 1B) (Page 14).

3. Figure 2. Unclear where the tissues were sampled from the stroke brains for the various assays. Not sure why A151 reduced the protein complexes of AIM2 and ASC, and caspase-1 at both mRNA and protein levels.

Thank you very much for the question, for mice that underwent MCAO treated with or without A151, the tissues were sampled from the infarct marginal zone of the left hemisphere to extract RNAs and proteins, and for the sham operated mice, the corresponding sites in the left cerebral hemisphere were collected.

Thank you very much for the question. Pyroptosis is closely related to inflammasome activation, is a programmed cell death mediated by caspase-1[11]. In addition to the post-translational assembly of inflammasome that is directly triggered by dsDNA, type I interferons stimulated by the cGAS-STING pathway has been shown to be capable of enhancing the AIM2 inflammasome response by providing inflammasome-priming signals [25,26,27], and promoting the expression of inflammasome platforms and substrates [25,28], in the present study, we thus further examined the expression of the protein complexes of AIM2 and ASC, and caspase-1 at both mRNA and protein levels following cGAS inhibition via A151.

4. Figure 3A. Not sure if any evidence for pyroptosis. The authors claimed that the GSDMD was found at cell plasma membrane; therefore this was consistent with pyroptosis. However, this is hardly a supporting evidence for such a conclusion. The images could not tell whether it's cytosolic or cell membrane staining. No evidence was provided to indicate whether these GSDMD positive Iba-1 cells were live or dead/dying.

Thank you very much for pointing this problem. GSDMD has been demonstrated as a pore-forming membrane protein during pyroptosis, forming a “ring of fire”. In addition, morphological evidence of caspase-1 and GSDMD immunoreactivity were consistent with that observed in the CNS in both MS and EAE [12], thus we claimed that the GSDMD was found at cell plasma membrane; therefore this was consistent with pyroptosis.

Given that GSDMD acts as a pore-forming membrane protein during pyroptosis, and pyroptosis is an inflammatory programmed cell death, GSDMD positive Iba-1 cells may indicate a form and a progress of cell death, these GSDMD positive Iba-1 cells may be damaged and dying cells [12].

We also performed double immunofluorescent double staining of MCAO mice brain at day 3 was performed to assess the localization of GSDMD (red) compared to microglia marker Iba-1 (green) in the ischemic brain (Fig. S9). This representative image illustrates that microglia undergoing pyroptosis (indicated by reference boxes A red, B yellow. and C green; defined by GSDMD immunoreactivity [red], and diminished Iba-1 expression [green]), the Iba-1 expression diminished with increasing GSDMD expression, healthy microglia that are GSDMD- are shown the highest Iba-1 expression. Right panels indicate results of a 2.5D intensity analysis of Iba-1 and GSDMD immunostaining (Page 16).

Figure S9

In vivo microglial pyroptosis stained for Iba-1 and GSDMD in brain section of MCAO mice.

References

- [1] M. Gelderblom, A. Weymar, C. Bernreuther, J. Velden, P. Arunachalam, K. Steinbach, E. Orthey, T.V. Arumugam, F. Leypoldt, O. Simova, Neutralization of the IL-17 axis diminishes neutrophil invasion and protects from ischemic stroke, *Blood* 120 (2012) 3793–3802.
- [2] N.D. Kim, A.D. Luster, The role of tissue resident cells in neutrophil recruitment, *Trends in immunology* 36 (2015) 547–555.
- [3] J.-K. Strecker, A. Schmidt, W.-R. Schäbitz, J. Minnerup, Neutrophil granulocytes in cerebral ischemia—evolution from killers to key players, *Neurochemistry international* 107 (2017) 117–126.
- [4] H. Ren, Y. Kong, Z. Liu, D. Zang, X. Yang, K. Wood, M. Li, Q. Liu, Selective NLRP3 (pyrin domain-containing protein 3) inflammasome inhibitor reduces brain injury after intracerebral hemorrhage, *Stroke* 49 (2018) 184–192.
- [5] D. Li, W. Lang, C. Zhou, C. Wu, F. Zhang, Q. Liu, S. Yang, J. Hao, Upregulation of microglial zeb1 ameliorates brain damage after acute ischemic stroke, *Cell reports* 22 (2018) 3574–3586.
- [6] C. Qin, W.-H. Fan, Q. Liu, K. Shang, M. Murugan, L.-J. Wu, W. Wang, D.-S. Tian, Fingolimod protects against ischemic white matter damage by modulating microglia toward M2 polarization via STAT3 pathway, *Stroke* 48 (2017) 3336–3346.
- [7] P.M. Wise, D.B. Dubal, M.E. Wilson, S.W. Rau, M. Böttner, K.L. Rosewell, Estradiol is a protective factor in the adult and aging brain: understanding of mechanisms derived from in vivo and in vitro studies, *Brain Research Reviews* 37 (2001) 313–319.
- [8] S.-H. Yang, J. Shi, A.L. Day, J.W. Simpkins, Estradiol exerts neuroprotective effects when administered after ischemic insult, *Stroke* 31 (2000) 745–748.
- [9] M. Li, Z. Li, Y. Yao, W.-N. Jin, K. Wood, Q. Liu, F.-D. Shi, J. Hao, Astrocyte-derived interleukin-15 exacerbates ischemic brain injury via propagation

- of cellular immunity, *Proceedings of the National Academy of Sciences* 114 (2017) E396–E405.
- [10] J. Barrington, E. Lemarchand, S.M. Allan, A brain in flame; do inflammasomes and pyroptosis influence stroke pathology?, *Brain Pathology* 27 (2017) 205–212.
- [11] P. Broz, V.M. Dixit, Inflammasomes: mechanism of assembly, regulation and signalling, *Nature Reviews Immunology* 16 (2016) 407.
- [12] B.A. McKenzie, M.K. Mamik, L.B. Saito, R. Boghazian, M.C. Monaco, E.O. Major, J.-Q. Lu, W.G. Branton, C. Power, Caspase-1 inhibition prevents glial inflammasome activation and pyroptosis in models of multiple sclerosis, *Proceedings of the National Academy of Sciences* 115 (2018) E6065–E6074.
- [13] Y. Fu, Q. Liu, J. Anrather, F.-D. Shi, Immune interventions in stroke, *Nature Reviews Neurology* 11 (2015) 524.
- [14] Y.-Y. Lu, Z.-Z. Li, D.-S. Jiang, L. Wang, Y. Zhang, K. Chen, X.-F. Zhang, Y. Liu, G.-C. Fan, Y. Chen, TRAF1 is a critical regulator of cerebral ischaemia-reperfusion injury and neuronal death, *Nature communications* 4 (2013) 2852.
- [15] R. Zhang, R.B. Diasio, Z. Lu, T. Liu, Z. Jiang, W.M. Galbraith, S. Agrawal, Pharmacokinetics and tissue distribution in rats of an oligodeoxynucleotide phosphorothioate (GEM 91) developed as a therapeutic agent for human immunodeficiency virus type-1, *Biochemical pharmacology* 49 (1995) 929–939.
- [16] B. Peng, J. Andrews, I. Nestorov, B. Brennan, P. Nicklin, M. Rowland, Tissue distribution and physiologically based pharmacokinetics of antisense phosphorothioate oligonucleotide ISIS 1082 in rat, *Antisense and Nucleic Acid Drug Development* 11 (2001) 15–27.
- [17] L. Dong, S. Ito, K.J. Ishii, D.M. Klinman, Suppressive oligodeoxynucleotides delay the onset of glomerulonephritis and prolong survival in lupus-prone NZB × NZW mice, *Arthritis & Rheumatism: Official Journal of the American College of Rheumatology* 52 (2005) 651–658.
- [18] F.R. Bahjat, R.L. Williams-Karnesky, S.G. Kohama, G.A. West, K.P. Doyle, M.D. Spector, T.R. Hobbs, M.P. Stenzel-Poore, Proof of concept: pharmacological preconditioning with a Toll-like receptor agonist protects against cerebrovascular injury in a primate model of stroke, *Journal of Cerebral Blood Flow & Metabolism* 31 (2011) 1229–1242.
- [19] H. Scholtzova, R.J. Kascsak, K.A. Bates, A. Boutajangout, D.J. Kerr, H.C. Meeker, P.D. Mehta, D.S. Spinner, T. Wisniewski, Induction of toll-like receptor 9 signaling as a method for ameliorating Alzheimer's disease-related pathology, *Journal of Neuroscience* 29 (2009) 1846–1854.
- [20] C. Lu, T. Ha, X. Wang, L. Liu, X. Zhang, E.O. Kimbrough, Z. Sha, M. Guan, J. Schweitzer, J. Kalbfleisch, The TLR 9 Ligand, CpG-ODN, Induces Protection against Cerebral Ischemia/Reperfusion Injury via Activation of

- PI3K/Akt Signaling, *Journal of the American Heart Association* 3 (2014) e000629.
- [21] L. Dong, S.i. Ito, K.J. Ishii, D.M. Klinman, Suppressive oligonucleotides protect against collagen-induced arthritis in mice, *Arthritis & Rheumatism* 50 (2004) 1686–1689.
- [22] P.P. Ho, P. Fontoura, P.J. Ruiz, L. Steinman, H. Garren, An immunomodulatory GpG oligonucleotide for the treatment of autoimmunity via the innate and adaptive immune systems, *The Journal of Immunology* 171 (2003) 4920–4926.
- [23] H. Shirota, I. Gursel, M. Gursel, D.M. Klinman, Suppressive oligodeoxynucleotides protect mice from lethal endotoxic shock, *The Journal of Immunology* 174 (2005) 4579–4583.
- [24] C. Fujimoto, D. Klinman, G. Shi, H. Yin, B. Vistica, J. Lovaas, E. Wawrousek, T. Igarashi, C.C. Chan, I. Gery, A suppressive oligodeoxynucleotide inhibits ocular inflammation, *Clinical & Experimental Immunology* 156 (2009) 528–534.
- [25] F. Martinon, A. Mayor, J. Tschopp, The inflammasomes: guardians of the body, *Annual review of immunology* 27 (2009) 229–265.
- [26] L.I. Labzin, M.A. Lauterbach, E. Latz, Interferons and inflammasomes: cooperation and counterregulation in disease, *Journal of Allergy and Clinical Immunology* 138 (2016) 37–46.
- [27] F. Liu, Q. Niu, X. Fan, C. Liu, J. Zhang, Z. Wei, W. Hou, T.-D. Kanneganti, M.L. Robb, J.H. Kim, Priming and activation of inflammasome by canarypox virus vector ALVAC via the cGAS/IFI16–STING–type I IFN pathway and AIM2 sensor, *The Journal of Immunology* 199 (2017) 3293–3305.
- [28] J. Lugrin, F. Martinon, The AIM 2 inflammasome: Sensor of pathogens and cellular perturbations, *Immunological reviews* 281 (2018) 99–114.

3rd Editorial Decision

11th February 2020

Thank you for the submission of your revised manuscript to EMBO Molecular Medicine. We have now received the enclosed report from the two referees who were asked to re-assess it. As you will see below that both referees think that while the majority of the concerns raised by all three referees have been addressed, several issues remain. I think that the referees' recommendations are rather clear and there is therefore no need to repeat their comments. All the remaining issues raised by the referees need to be satisfactorily addressed.

***** Reviewer's comments *****

Referee #1 (Comments on Novelty/Model System for Author):

Large numbers of experiments and combination of in vivo and in vitro assays lend strength. Think it is worthy of publication if the authors address the remaining issues.

Referee #1 (Remarks for Author):

The authors present a revised manuscript on the effect of cGAS inhibition on pyroptosis,

inflammation, and outcomes in a murine model of ischemic stroke. Overall, the authors were responsive to the initial review and the manuscript is improved. In particular, the addition of aging mice, longer term outcomes, improved Western blots, primary microglia, and confirmation of microglial-specific cGAS KO in the inducible Cre system add significant support for their conclusions.

1. The aged mice need to be added to the mouse section of the methods and age of the aged mice is reported in the response to reviewers but not the manuscript. The authors should also consider that 8 week mice are equivalent to young adults and 12 month mice equivalent to middle aged. A helpful website is <https://www.jax.org/news-and-insights/jax-blog/2017/november/when-are-mice-considered-old> The addition of the 12 month mice is a strength but they should be referred to as older or aging, not aged.
2. The timing of initiation of A151 treatment (at reperfusion?) needs to be stated in methods.
3. Representative western in Figure 2E shows increased ASC in MCAO+A151, not decreased as is shown in the summary figure 2F. Please explain. Presenting individual data points rather than bar charts will provide the reader with a much more informed view of the data and variability.

Referee #2 (Comments on Novelty/Model System for Author):

The authors have performed a fair amount of extra work to strengthen their conclusions. Using the present tools (i.e. no in vivo imaging to look at microglia/neuronal responses, or further clarification on the molecular mechanisms, etc) the paper is not providing high level of mechanistic insight on how exactly cGAS contributes to brain injury, apart from the fact that A151 is effective to dampen inflammation and improve functional outcome after experimental stroke. This observation could be still interesting enough for the field.

Referee #2 (Remarks for Author):

The authors have responded to most points raised and performed additional experiments.

Minor point, Page 17: „the leakage of dsDNA into the cytoplasm was also detected at time points of 6 h, 24 h, as well as 3 days after MCAO (Fig.1A)". As no staining was used to visualize the nuclear membrane, it would be more precise referring to disintegration of the nucleus instead.

2nd Revision - authors' response

23rd February 2020

Thank you very much for your kind letter and thank you for taking your time and effort to review our article entitled “Inhibition of double-strand DNA sensing cGAS ameliorates brain injury after ischemic stroke” (Manuscript No.: EMM-2019-11002-V4). We really appreciate all your comments and suggestions. These comments and suggestions are all very valuable and have enabled us to improve our article. We have carefully revised the manuscript according to the reviewer’s comments and made itemized responses below. In this revised version, we have uploaded a copy of the original manuscript with all the changes highlighted by using red colored text. We appended our point-by-point response to the comments raised by the reviewers to this letter.

***** Reviewer's comments *****

Referee #1 (Comments on Novelty/Model System for Author):

Large numbers of experiments and combination of in vivo and in vitro assays lend strength. Think it is worthy of publication if the authors address the remaining issues.

Referee #1 (Remarks for Author):

The authors present a revised manuscript on the effect of cGAS inhibition on pyroptosis, inflammation, and outcomes in a murine model of ischemic stroke. Overall, the authors were responsive to the initial review and the manuscript is improved. In particular, the addition of aging mice, longer term outcomes, improved Western blots, primary microglia, and confirmation of microglial-specific cGAS KO in the inducible Cre system add significant support for their conclusions.

1. The aged mice need to be added to the mouse section of the methods and age of the aged mice is reported in the response to reviewers but not the manuscript. The authors should also consider that 8 week mice are equivalent to young adults and 12 month mice equivalent to middle aged. A helpful website is <https://www.jax.org/news-and-insights/jax-blog/2017/november/when-are-mice-considered-old> The addition of the 12 month mice is a strength but they should be referred to as older or aging, not aged.

Thank you very much for the suggestion. We have mentioned this in the method section as Eight weeks old and 12 month old male mice were used in this study (Page 5). And for 12 month mice, we have modified aged mice as older mice throughout the manuscript, thank you very much for pointing out this and nicely providing the useful website.

2. The timing of initiation of A151 treatment (at reperfusion?) needs to be stated in methods.

Thank you very much for the suggestion, the timing of initiation of A151 treatment starts immediately after immediately after reperfusion, and we have mentioned this in the method section (Page 6).

3. Representative western in Figure 2E shows increased ASC in MCAO+A151, not decreased as is shown in the summary figure 2F. Please explain. Presenting individual data points rather than bar charts will provide the reader with a much more informed view of the data and variability.

Thank you very much for pointing out this, we are very sorry for the confused we made. We checked the data and found that we presented a wrong picture as shown, and we have replaced with the right representative western blotting of ASC, and the ASC protein level was reduced in MCAO+A151, individual data points are shown.

Referee #2 (Comments on Novelty/Model System for Author):

The authors have performed a fair amount of extra work to strengthen their conclusions. Using the present tools (i.e. no in vivo imaging to look at microglia/neuronal responses, or further clarification on the molecular mechanisms, etc) the paper is not providing high level of mechanistic insight on how exactly cGAS contributes to brain injury, apart from that fact that A151 is effective to dampen inflammation and improve functional outcome after experimental stroke. This observation could be still interesting enough for the field.

Referee #2 (Remarks for Author):

The authors have responded to most points raised and performed additional experiments.

Minor point, Page 17: „the leakage of dsDNA into the cytoplasm was also detected at time points of 6 h, 24 h, as well as 3 dafter MCAO (Fig.1A)". As no staining was used to visualize the nuclear membrane, it would be more precise reffering to disintegration of the nucleus instead.

We are very grateful for the suggestion, to be more precise, we have modified the sentence according to this previous comment, and modified “the leakage of dsDNA into the cytoplasm” as “disintegration of the nucleus” (Page 14).

Accepted

25th February 2020

We are pleased to inform you that your manuscript is accepted for publication and is now being sent to our publisher to be included in the next available issue of EMBO Molecular Medicine.

Corresponding Author Name: Lihua Wang and Junwei Hao

Journal Submitted to: Embo molecular medicine

Manuscript Number: EMM-2019-11002-V2-Q